# Comparison of south-east Atlantic aerosol direct radiative effect over clouds from SCIAMACHY, POLDER and OMI-MODIS

**Martin de Graaf**[1], **Ruben Schulte**[2], **Fanny Peers**[3], **Fabien Waquet**[4], **L. Gijsbert Tilstra**[1], **and Piet Stammes**[1]

[1]Satellite Observations Department, Royal Netherlands Meteorological Institute (KNMI), De Bilt, The Netherlands
[2]Geosciences & Remote Sensing Department, Delft University of Technology (TUD), Delft, The Netherlands
[3]University of Exeter, Exeter, United Kingdom
[4]Université des Sciences et Technologies de Lille 1, Lille, France

**Correspondence:** Martin de Graaf (martin.de.graaf@knmi.nl)

**Abstract.** The Direct Radiative Effect (DRE) of aerosols above clouds has been found to be significant over the south-east Atlantic Ocean during the African biomass burning season due to elevated smoke layers absorbing radiation above the cloud deck. So far, global climate models have been unsuccessful in reproducing the high DRE values measured by various satellite instruments. Meanwhile, the radiative effects by aerosols have been identified as the largest source of uncertainty in global climate models. In this paper, three independent satellite datasets of DRE during the biomass burning season in 2006 are compared to constrain the south-east Atlantic radiation budget. The DRE of aerosols above clouds is derived from the spectrometer Scanning Imaging Absorption Spectrometer for Atmospheric CHartographY (SCIA-MACHY), the polarimeter Polarization and Directionality of the Earth's Reflectances (POLDER), and from collocated measurements by the spectrometer Ozone Monitoring Instrument (OMI) and the imager Moderate Resolution Imaging Spectroradiometer(MODIS). All three datasets confirm the high DRE values during the biomass season, underlining the relevance of local aerosol effects. Differences between the instruments can be attributed mainly to sampling issues. When these are accounted for, the remaining differences can be explained by a higher cloud optical thickness (COT) derived from POLDER compared to the other instruments, and a neglect of aerosol optical thickness (AOT) at shortwave infrared (SWIR) wavelengths in the method used for SCIAMACHY and OMI-MODIS. The higher COT from POLDER by itself can explain the difference found in DRE between POLDER and the other instruments. The AOT underestimation is mainly evident at high values of the aerosol DRE and accounts for about a third of the difference between POLDER and OMI-MODIS DRE. The datasets from POLDER and OMI-MODIS effectively provide lower and upper bound for the aerosol DRE over clouds over the south-east Atlantic, which can be used to challenge Global Circulation Models (GCMs). Comparisons of model and satellite datasets should also account for sampling issues. The complementary DRE retrievals from OMI-MODIS and POLDER may benefit from upcoming satellite missions that combine spectrometer and polarimeter measurements.

## 1 Introduction

During the monsoon dry season in Africa, biomass burning from wildfires produces huge amounts of carbonaceous aerosols, or smoke (de Graaf et al., 2010). The smoke that is transported over the south-east Atlantic Ocean overlies one of the planet's major stratocumulus cloud decks (Swap et al., 1996). Smoke is a light-absorbing aerosol and the instantaneous change in radiative flux by the scattering and absorption of sunlight is known as the aerosol direct radiative effect (DRE). The absorption of sunlight by aerosols adds heat to the atmosphere at the aerosol layer height, changing the atmospheric stability and the amount of radiation received at the surface (Yu et al., 2002), which in turn affects the development of clouds (Feingold et al., 2005) and precipitation (Sorooshian et al., 2009). Absorbing aerosols in or near clouds may evaporate cloud droplets (Ackerman et al., 2000), while absorbing aerosols above marine stratocumulus

clouds may increase the temperature inversion, thickening the cloud (Johnson et al., 2004; Wilcox, 2010). These rapid adjustments to radiative flux changes are known as aerosol semi-direct climate effects. Furthermore, aerosols impact the formation of clouds by acting as cloud condensation nuclei, known as the aerosol indirect effect. Aerosol climate impacts are expected to counteract a significant part of the greenhouse gas-induced global warming, which is estimated at $+2.8\pm0.3$ W m$^{-2}$, but the large uncertainty of aerosol-radiation interactions, ranging from 0 to -0.9 W m$^{-2}$, limits our ability to attribute climate change and improve the accuracy of climate change projections (Boucher et al., 2013).

Constraining aerosol effects in model studies remains a challenge as observations of aerosol direct, indirect, and semi-direct effects are scarce. The main problems are the complexities involved in untangling the observations of aerosols, clouds and radiation in the real world. In this paper, we focus on the direct effect of aerosols above clouds, which can be characterized relatively well due to recent developments in retrieval techniques from a number of different satellite instruments.

The radiative effect of an atmospheric constituent can be defined as the net broadband irradiance change $\Delta F$ at a certain level with and without the forcing constituent, after allowing for stratospheric temperatures to readjust to radiative equilibrium, but with tropospheric and surface temperatures and state held fixed at the unperturbed values (Forster et al., 2007). For tropospheric aerosols as the forcing agent, stratospheric adjustments have little effect on the radiative effect and the instantaneous irradiance change at the Top Of the Atmosphere (TOA) can be substituted. The instantaneous aerosol direct radiative effect at TOA is therefore defined as the change in net (upwelling minus downwelling) irradiance, due to the introduction of aerosols in the atmosphere. Since at TOA the downwelling irradiance $F^{\downarrow}$ is the incoming solar irradiance $F_0$ for all scenes, the aerosol DRE for a cloud scene can be determined from the difference between the upwelling irradiance in an aerosol-free cloud scene $F^{\uparrow}_{\mathrm{cld}}$ and the upwelling irradiance of a scene with the same clouds plus aerosols $F^{\uparrow}_{\mathrm{cld+aer}}$ :

$$\mathrm{DRE_{aer}} = (F^{\downarrow}-F^{\uparrow})_{\mathrm{cld}}-(F^{\downarrow}-F^{\uparrow})_{\mathrm{cld+aer}} = F^{\uparrow}_{\mathrm{cld+aer}}-F^{\uparrow}_{\mathrm{cld}}. \tag{1}$$

A radiative transfer model (RTM) is commonly used to, given the atmospheric constituents in the atmosphere, simulate the scene twice; once with and once without the aerosols. To do this for a scene with aerosols overlying a cloud, the optical and physical properties of both the aerosols and the clouds have to be determined, and to a lesser extend the light absorption and scattering properties of the air and the surface reflectance.

The DRE (at TOA) due to the light absorbing species in smoke is strongly affected by the presence of clouds. Over the dark ocean, in cloud-free scenes, the upwelling radiation at TOA is dominated by the scattering from aerosols and the planetary albedo is increased by the presence of aerosols, resulting in a negative direct effect (cooling). Over clouds, on the other hand, scattering by aerosols hardly contribute to the upwelling radiation at TOA, since the scattering by clouds is dominant. However, the aerosols absorb radiation, lowering the planetary albedo, resulting in a positive direct effect (warming). E.g. an average change in forcing efficiency (DRE divided by AOT) from $-25$ W m$^{-2}\tau^{-1}$ in cloud-free scenes to $+50$ W m$^{-2}\tau^{-1}$ in fully clouded scenes was found by Chand et al. (2009). The DRE changed sign at a critical cloud fraction of about $0.4$ for scenes over the south-east Atlantic Ocean. Similarly, simulations show that the DRE changed sign at a critical cloud optical thickness (COT) of about 4–8, a higher COT resulting in a higher DRE (Feng and Christopher, 2015).

The south-east Atlantic has been a strong focus of modeling and observational studies of the aerosol DRE over clouds. The ocean west of the African continent, where sea surface temperatures are low due to upwelling of cold deep sea water, is covered by a semi-permanent cloud deck. During the austral winter months (July – October), which is the dry season on the adjacent African continent, a myriad of vegetation fires produces immense amounts of smoke ($\sim 25$ Tg black carbon per year), resulting in the largest source of black carbon and natural carbonaceous species in the atmosphere worldwide (van der Werf et al., 2010).

The combination of large areas of boundary layer clouds and overlying smoke proved to be a huge challenge for GCMs to simulate consistent aerosol DRE values at TOA. A comparison of sixteen GCMs showed a large range of aerosol DRE over the south-east Atlantic, from strongly negative (cooling) to strongly positive (warming) for the same experiment (Zuidema et al., 2016), depending on the models' details on cloud and aerosol microphysical properties. It also shows that aerosol radiative effects can be very important on the local scale, near the source areas, even if the contribution to the global radiative budget can be small.

Observations are needed to constrain the model simulations. This can be challenging, because ground observations are sparse and scarce, and satellite observations of COT and aerosol optical thickness (AOT) are difficult to disentangle. Satellite COT observations in the common visible spectral region are biased by absorption by aerosols, resulting in a biased DRE estimation (Haywood et al., 2004; Coddington et al., 2010). Satellite AOT retrievals are commonly performed only in cloud-free scenes, hampering the computation of the aerosol DRE in cloud scenes.

One way of separating cloud and aerosol scattering is the use of active (lidar) instruments, which produce vertically high resolution backscatter profiles, e.g. CALIOP onboard CALIPSO (Chand et al., 2009; Meyer et al., 2013; Zhang et al., 2014). Unfortunately, the spatial coverage of a lidar is limited. Another solution is the use of polarime-

ter measurements. The different effects of spherical water droplets and irregularly shaped aerosol particles on the polarization of light can be used to separate the cloud and aerosol contribution to the radiation at TOA. This was applied to POLDER measurements (Waquet et al., 2013a). The absorption from the aerosol layer and the COT is retrieved using reflectances measured in the visible and shortwave infrared. Knowing the COT, and AOT of overlying aerosols, the aerosol DRE in cloud scenes can be computed using an RTM twice, simulating the upwelling radiation for the cloud scene with ($F^{\uparrow}_{\mathrm{cld+aer}}$) and without the aerosols ($F^{\uparrow}_{\mathrm{cld}}$). The monthly averaged instantaneous DRE values from POLDER for aerosols over clouds over the south-east Atlantic Ocean in August 2006 found in this way was 33 W m$^{-2}$ (Peers et al., 2015).

The absorption by small smoke aerosols is especially strong in the UV. Several methods use this principle to separate the cloud scattering from the aerosol absorption and scattering. The strong UV absorption can be quantified by the Absorbing Aerosol Index (AAI) (de Graaf et al., 2005, 2007; Wilcox, 2012), while the reduction in reflectance in the UV and visible channels can be simulated using LookUp Tables (LUTs). In this way, the AOT of smoke above clouds was retrieved over the south-east Atlantic, with the COT of the clouds underneath retrieved simultaneously, using OMI measurements (Torres et al., 2011). A similar method was applied to MODIS measurements to retrieve AOT and COT simultaneously, using measurements in the visible (Jethva et al., 2013).

These methods all rely on the quantification of the optical properties of the aerosols. However, light absorption by smoke is highly variable and the spectral dependence (quantified by the Ångström exponent) is much larger than often assumed (Jethva and Torres, 2011) and not necessarily unique (Bergstrom et al., 2007). The AOT over clouds in the south-east Atlantic derived from POLDER, CALIOP and MODIS measurements were compared in Jethva et al. (2014), showing a general agreement, but large differences in the details.

Spectral information of the aerosol and cloud properties is needed to correctly specify the aerosol-cloud-radiation interactions at all wavelengths. Measurements from six wavelength channels from MODIS (from 0.47–1.24$\mu$m) have been used to retrieve COT and cloud droplet effective radius (CER) for clouds with overlying aerosols, simultaneously with the above–cloud AOT, and subsequently aerosol DRE (Meyer et al., 2015). However, here also the aerosol spectral properties have to be assumed. To circumvent the use of aerosol optical property models altogether, the spectral dependence of aerosol absorption can be measured with hyperspectral satellite instruments like SCIAMACHY (de Graaf et al., 2012). The principle here is that the absorption by the aerosols is captured entirely by the radiance measurements at TOA in the UV, visible and SWIR spectral regions (measured $F^{\uparrow}_{\mathrm{cld+aer}}$), and only the aerosol-free atmosphere is simulated in an RTM (simulated $F^{\uparrow}_{\mathrm{cld}}$). The cloud properties can be retrieved in the SWIR where small particles like smoke have little to no effect on the COT and CER. The DRE is then retrieved from a difference in simulated and measured reflectance, and the difference is attributed to absorption by aerosols. Hence it is termed differential aerosol absorption (DAA) method. The monthly averaged instantaneous DRE values from SCIAMACHY for aerosols over clouds over the south-east Atlantic in August 2006 found in this way was 23 W m$^{-2}$. The DRE from SCIAMACHY was compared to Hadley Centre Global Environmental Model version 2 (HadGEM2) climate model simulations (de Graaf et al., 2014). Simulated monthly averaged aerosol DRE from HadGEM2 were a factor of 5 lower than SCIAMACHY observations, showing that even this GCM, which simulated a large warming over the south-east Atlantic, still fell short in simulating the UV-absorption by smoke. The DAA method was recently applied to a combination of OMI and MODIS reflectance measurements. The monthly averaged instantaneous DRE values from OMI-MODIS for aerosols over clouds over the south-east Atlantic in August 2006 was 25 W m$^{-2}$ (de Graaf et al., 2019).

The main challenge in comparing satellite data is the wide range in spatial resolution and sampling of different instruments. To resolve this, many papers report area- and time-averaged DRE values and compare them to other average values of the aerosol DRE. In this paper, the DRE derived from POLDER measurements are compared to the DRE from SCIAMACHY and to DRE derived from a combination of OMI and MODIS measurements, accounting explicitly for sampling issues. POLDER reports consistently high values of AOT, COT and DRE compared to other instruments, and we show that the DRE values agree to within the uncertainty estimates when sampling issues are accounted for and the differences in AOT and COT with other instruments are taken into account.

## 2 Methods

### 2.1 POLDER DRE

POLDER is a passive optical imaging radiometer and polarimeter on-board the Polarization and Anisotropy of Reflectances for Atmospheric Science coupled with Observations from a Lidar (PARASOL) (Deschamps et al., 1994). PARASOL was launched in December 2004 and was part of the A-Train satellite constellation for five years. After 2009, PARASOL's orbit was lowered, and it fully exited the A-Train in 2013. POLDER provides radiances in nine spectral bands between 443 and 1020 nm and polarization measurements at 490, 670 and 865 nm. The ground spatial resolution is about $5.3 \times 6.2$ km$^2$ and the swath width about 1100 km (Deschamps et al., 1994). All measurements of POLDER are projected on a fixed global reference grid of 6×6 km$^2$.

**Table 1.** Spatial and temporal resolution of the different satellite instruments as used in this paper. Grid sizes of SCIAMACHY and OMI are those at nadir, grid sizes of POLDER and MODIS are fixed.

| Instrument | Platform | Local equator crossing time | Global coverage (days) | Pixel size (km $\times$ km) | Operation period |
|---|---|---|---|---|---|
| POLDER | PARASOL | 13:33 | 1 | $6 \times 6$ | 2004 – 2013 |
| SCIAMACHY | EnviSat | 10:00 | 6 | $60 \times 30$ | 2002 – 2012 |
| OMI | Aura | 13:38 | 1 | $13 \times 24$ | 2004 – present |
| MODIS | Aqua | 13:30 | 1 | $0.5 \times 0.5$ | 2002 – present |

Using POLDER measurements, the above-cloud AOT, the aerosol Single Scattering Albedo (SSA) and the COT are retrieved in two steps. The first one consists of using the polarization radiance measurements to retrieve the scattering AOT and the aerosol size distribution in a cloudy scene. Aerosols affect the polarization in a cloudy scene in two ways. Firstly, the large peak of the signal around a scattering angle of $140°$, caused by the liquid cloud droplets, is attenuated. Secondly, an additional signal at side scattering angles is created. The effect of absorption is assumed to be very weak at these angles and mostly treated as a scattering process. In the second step, the spectral contrast and the magnitude of the total radiances measured in the visible and SWIR are used to retrieve the absorption AOT and COT simultaneously. Therefore, the retrieval of the aerosol properties is done with minimal assumptions and with the cloud properties corrected for the overlying aerosol absorption. To ensure the quality of the products, several filters are applied, which include the removal of inhomogeneous clouds, broken clouds, cloud edges, clouds with COT lower than 3 and cirrus (Waquet et al., 2013b; Peers et al., 2015).

The POLDER DRE is finally calculated over the south-east Atlantic for aerosols over clouds in 2006 using the retrieved AOT, SSA and COT with the method described in section 3 of Peers et al. (2015). POLDER apparent $O_2$ cloud top pressures were used to constrain the cloud layer height, although the cloud top pressure has been shown to have a negligible effect on the TOA radiation (less than 1 % for a change of 200 hPa (Ahmad et al., 2004; de Graaf et al., 2012)). CER was derived from collocated MODIS measurements. CER may be retrieved directly by POLDER for specific cases using the separation of the peaks in the polarized scattering phase function (Bréon and Doutriaux-Boucher, 2005), but here we use the data as described in (Peers et al., 2015).

The DRE was derived for all scenes with a geometric cloud fraction (CF) of 1.0 and a COT larger than 3.0. The surface reflectance was computed taking surface winds into account (Cox and Munk, 1954), but since only scenes with a minimum COT of 3 were used, the influence of the surface reflectance on the total radiation field will be small. The ozone and the water vapor content were obtained from meteorological reanalysis.

## 2.2 SCIAMACHY DRE

The DAA method was developed for reflectance spectra from the SCanning Imaging Absorption spectroMeter for Atmospheric CHartographY (SCIAMACHY). SCIAMACHY was part of the payload of the Environment Satellite (EnviSat), launched in 2002, into a polar orbit with an equator crossing time of 10:00 local solar time in a descending (southward) direction, but stopped delivering data in 2012. SCIAMACHY observed radiation in two alternating modes, nadir and limb, yielding data blocks called states, approximately $960 \times 490$ km$^2$ in size. In nadir mode, SCIAMACHY measured continuous reflectance spectra from 240–2380 nm with a spatial resolution of about $60 \times 30$ km$^2$ and a spectral resolution of 0.2–1.5 nm (Bovensmann et al., 1999). This unique spectral range from the UV to the shortwave infrared (SWIR) contains 92 % of the incoming solar irradiance. The DRE was determined from SCIAMACHY reflectance spectra of cloud scenes in 2006 over the south-east Atlantic. Cloud properties were determined at 1.2 and 1.6 $\mu$m, where absorption by smoke is assumed to be negligible. Effective CF and cloud pressure (CP) were determined from (FRESCO) $O_2$-A band retrievals (Wang et al., 2008). All scenes with effective CF $> 0.3$, CP $> 850$ hPa and COT $> 3.0$ were used to select pixels with sufficient water clouds only. The ocean surface albedo was assumed to have a small, spectrally dependent, constant value. Total ozone was accounted for, but this has a negligible impact on the DRE. See de Graaf et al. (2012) for details.

## 2.3 DRE from combined OMI-MODIS reflectances

The absorption of radiation by aerosols is spectrally dependent, but since the particles vary in size and composition, the spectral dependence is smooth, as opposed to absorption by (trace) gases, which is strongly peaked in absorption lines. Therefore, the DRE data record from SCIAMACHY was continued using a combination of spectrally high-resolution OMI reflectances and low-resolution MODIS reflectances, which are sufficient to capture the spectral dependence of the absorption in the visible and SWIR.

OMI (Levelt et al., 2006), on-board the Aura satellite, was launched in 2004 in a polar orbit, crossing the equator around 13:30 local solar time in an ascending (northward) direction, to measure the complete spectrum from the UV to the visible

wavelength range (up to 500 nm) with a high spatial resolution, similar to SCIAMACHY. The Earth shine radiance is observed in a swath width of about 2600 km, covering almost the entire Earth in one day. The spatial resolution of OMI is typically about $15 \times 23.5$ km$^2$ at nadir to about $42 \times 126$ km$^2$ for far off-nadir (56 degrees) pixels. Since 2008, OMI suffers from progressive degradation, especially in far off-nadir pixels, called the row anomaly.

MODIS, on-board the Aqua satellite, flies in formation with Aura in the A-Train, leading Aura by about 15 minutes (in 2006, while PARASOL was placed in between these two instruments). MODIS measures radiances in broad bands (typical about 20–50 nm) from the visible to SWIR, with a typical spatial resolution of 250–500 m. Spectrally, OMI overlaps with MODIS at 459–479 nm (central wavelength 469 nm), which can be used to match the OMI reflectances in the visible channel and the MODIS reflectance in band 3 (de Graaf et al., 2016). This way, a continuous low-resolution spectrum at OMI resolution is available to which DAA can be applied (de Graaf et al., 2019).

The DRE was determined from OMI pixels over the southeast Atlantic in 2006. COT and CER were determined at 1.2 and 2.1 $\mu$m, because of a reduced sampling in MODIS/Aqua 1.6 $\mu$m band due to nonfunctional detectors (Meyer et al., 2015). CP and effective CF are available from OMI O$_2$-O$_2$ retrievals. All scenes with COT $> 3.0$, effective CF $> 0.3$ and CP $> 850$ hPa were selected. The ocean surface albedo was assumed to have a small, spectrally dependent, constant value.

The temporal and spatial resolutions of the various instruments compared in this paper are summarized in Table 1.

## 2.4 Error budget

The largest uncertainty for the DRE derives from the assumption that the aerosol-free cloud scene can be simulated using an RTM, which is assumed in all methods. For SCIAMACHY and OMI-MODIS scenes this was actually tested, by applying the technique to measured aerosol-free cloud scenes and determining the DRE, which should be zero by definition. This provides an easy verification of the method. For each instrument and area this can be determined separately, by screening cloud scenes with overlying absorbing aerosol using the AAI, which is highly sensitive to UV-absorbing aerosols. The (average) deviation of the DRE from zero, determined for aerosol-free cloud scenes, is a good estimate of the uncertainty of the method, which can be substantial. Such estimates are rarely given in the literature.

The dependence on uncertainties in the spectral properties of the overlying aerosols is small by DAA, because in this method the spectral measurements are used, not a model. Other minor error sources for the DAA method are the uncertainty in input parameters; the influence of the smoke on the estimated cloud fraction, cloud optical thickness and cloud droplet effective radius; an uncertainty in

**Table 2.** Maximum and average values of OMI-MODIS, SCIA-MACHY, and POLDER DRE on 12 and 19 August 2006 for the areas shown in Figures 1(d–f) and 1(a–c).

| 12 August 2006 | Max DRE | $\langle$DRE$\rangle$ |
|---|---|---|
| POLDER | 303.8 | 109.1 |
| OMI-MODIS | 120.0 | 35.5 |
| SCIAMACHY | 112.5 | 28.4 |
| 19 August 2006 | | |
| POLDER | 190.3 | 43.0 |
| OMI-MODIS | 94.0 | 11.4 |
| SCIAMACHY | 71.3 | 18.1 |

the anisotropy factor (de Graaf et al., 2019); and the uncertainty of estimating the COT and CER at SWIR wavelengths. The error on the aerosol DRE from SCIAMACHY is about 8 W m$^{-2}$ (de Graaf et al., 2012) and from OMI-MODIS about 13 W m$^{-2}$ (de Graaf et al., 2019).

The main source of error for the aerosol DRE over clouds from POLDER is the assumption on the aerosol refractive index. In the first step of the algorithm, an assumption on the refractive index is used in order to retrieve the above-cloud scattering AOT. In the second step, the imaginary part is modified in order to retrieve the absorption AOT from total reflectances, assuming the same real part of the refractive index as in the first step. The impact of the refractive index assumption on the DRE has been analyzed in (Peers et al., 2015) and a maximum error of 10 W m$^{-2}$ has been observed. Finally, an error on the CER can cause a bias of up to 10 % on the COT.

## 3 Results

### 3.1 Case studies in August 2006

The aerosol DRE retrievals over clouds from the various satellite instruments are first introduced in Figure 1 using two cases in August 2006, on the 12$^{\text{th}}$ and the 19$^{\text{th}}$. The first case shows the situation during the largest difference between the datasets, the second case the situation one week later, when the differences are moderate. Figures 1a–c show the same data as Figures 1d–f, only one week earlier and from a slightly different area, centered on the MERIS and SCIAMACHY overpass. Figures 1a and d show the POLDER DRE overlaid over a MODIS RGB image acquired around 13:10–13:20 UTC, Figures 1b and e show the OMI-MODIS DRE over the same MODIS RGB image, and Figures 1c and f the SCIAMACHY DRE overlaid over a MERIS RGB image, both on EnviSat. Envisat is in a morning orbit, and the SCIAMACHY and MERIS measurements were taken around 9:30–9:45 UTC. The clouds are more extensive in the latter image, because clouds in this area break

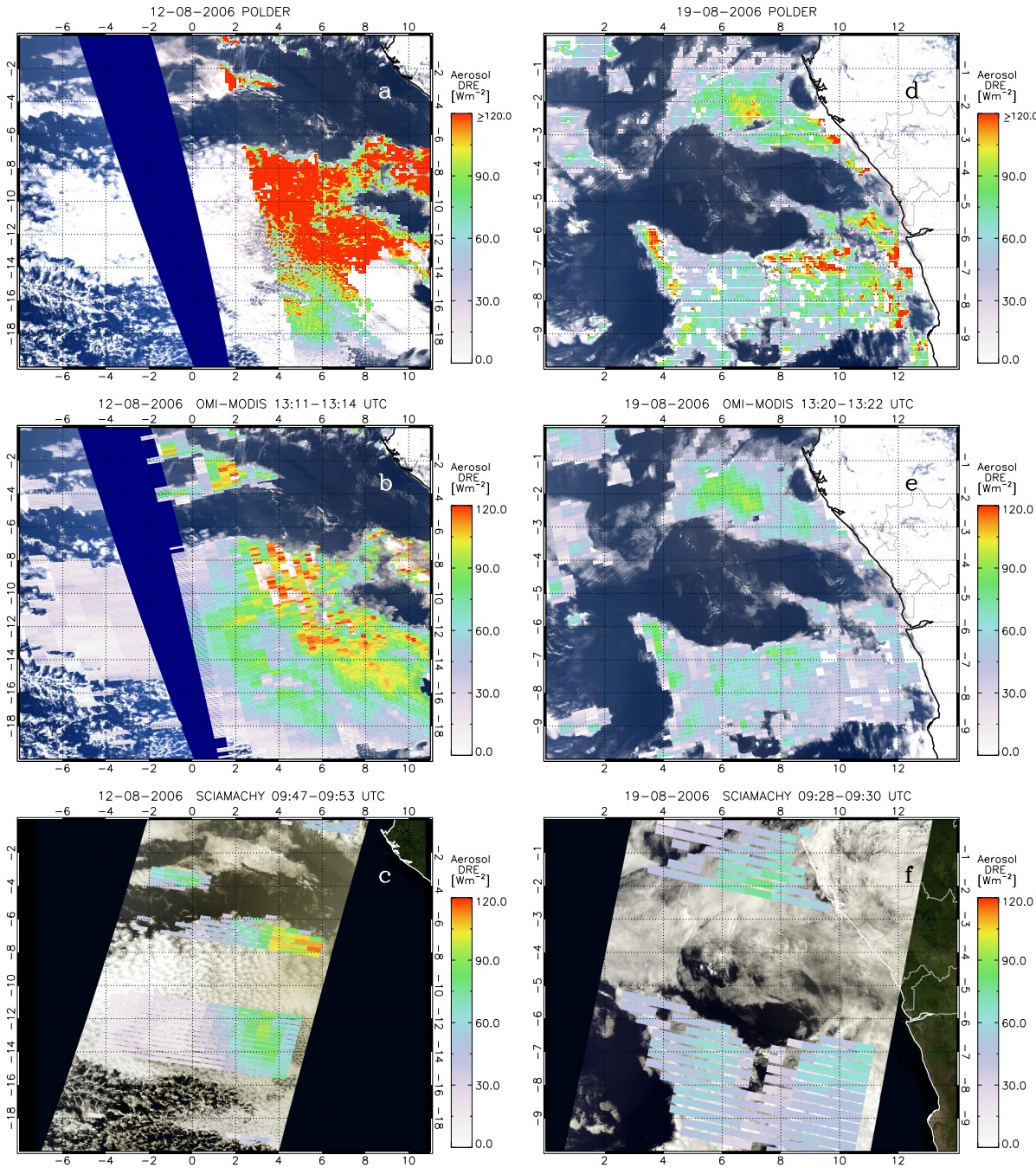

**Figure 1.** (a) Instantaneous Aerosol Direct Radiative Effect (DRE) over clouds on 12 August 2006 from POLDER, overlaid over a MODIS RGB image; (b) Aerosol DRE over clouds on the same day from a combination of OMI and MODIS reflectances, overlaid over the same MODIS RGB image; (c) Aerosol DRE over clouds from SCIAMACHY on the same day, overlaid over a MERIS RGB image; (d–f) same as (a–c) for 19 August 2006. The areas are centered over the MERIS/SCIAMACHY overpasses.

up as the day progresses and the solar radiation intensifies (Bergman and Salby, 1996).

During 2006 all instruments performed well, and August is the peak of the biomass burning season in southern Africa. An extended smoke plume, originating from the African con-

tinent, drifted over the south-east Atlantic Ocean in an elevated layer above a stratocumulus deck in the boundary layer. The absorption of radiation by the smoke above the stratocumulus cloud deck is indicated by high DRE values, in cloud scenes only.

Obviously, the spatial coverage of SCIAMACHY is much lower than OMI and MODIS, measuring in nadir mode only half of the time, and having larger pixels. Consequently, the OMI-MODIS DRE is smoother with a better coverage. However, the most striking feature is the much higher values from POLDER compared to the other two instruments, even though the general DRE patterns for the three instruments are quite similar. On 12 August 2006, the POLDER DRE is very large, reaching values up to 304 W m$^{-2}$. The OMI-MODIS DRE reaches up to 120 W m$^{-2}$, much lower than the POLDER DRE. The maximum SCIAMACHY DRE was 113 W m$^{-2}$. On 19 August 2006 the differences are smaller, but still obvious. The POLDER DRE reaches values up to 190 W m$^{-2}$ in parts where smoke from the African continent is abundant. The values drop off to zero over clouds where the smoke plume is thinning. The DRE from the two other instruments, on the other hand, is never larger than 100 W m$^{-2}$. The DRE values for these cases are summarized in Table 2.

Furthermore, a much higher area-averaged POLDER DRE on 12 August 2006 is found then for the other instruments, which is not only due to higher individual values. Figures 1d and 1e show that due to a smaller swath compared to OMI and MODIS, POLDER samples an area near the continent that has by coincidence only very high DRE values; the entire left part of the area in Fig. 1a is not sampled. This yields a much higher area-averaged DRE for POLDER than from the other instruments. OMI and MODIS sample the entire basin, where large parts have only very low to zero aerosol DRE values. In the case of SCIAMACHY only about 1/6 of the area is covered by SCIAMACHY nadir measurements, which obviously makes it very sensitive to the sampling of an aerosol plume during one overpass.

Additionally, the SCIAMACHY and OMI large pixel sizes smooth the high DRE values that are found by POLDER. Pixel sizes from SCIAMACHY are about 50 times as large as those from POLDER, which will result in a smoothing of small scale features, like local high values. OMI pixel sizes vary between nadir and the far off-nadir, being about 10 times larger than POLDER at nadir, up to 147 times larger at a viewing zenith angle of 56 degrees.

Lastly, on 12 August 2006 at some places where the highest values of aerosol DRE can be expected, the OMI-MODIS retrievals failed (Figure 1d), probably due to broken cloud scenes in combination with very high aerosol loadings, which resulted in low scene reflectances which were not marked as clouded scenes. Furthermore, cloud filtering can be different for the three instruments, due to the use of different cloud filters (use of effective or geometrical cloud fractions), which may have a strong influence on the (average) DRE.

The differences between the datasets will be explained below by a closer inspection of the data and the retrievals.

## 3.2 Area-averaged DRE

Clearly, sampling is an issue that needs to be considered when comparing datasets, and datasets and simulations. Often, area- and time-averages are compared, to reduce the effects of sampling differences. Here we show the effect of ignoring the sampling differences between instruments.

In Figure 2a, the area-averaged instantaneous aerosol DRE over clouds from all three instruments is given for all available data in the area 10°N–20°S,10°W–20°E, between 1 June and 1 October 2006. This is the biomass burning season and the area where often area-averaged DRE values have been reported during this season (e.g. Chand et al., 2009; de Graaf et al., 2014; Meyer et al., 2013; Peers et al., 2015). Since the instruments have different overpass times, the instantaneous aerosol DRE over clouds was normalized by dividing by the cosine of the solar zenith angle. Therefore, the quantity in Figure 2a represents the instantaneous aerosol DRE for an overhead Sun (at noon), which is generally higher than the instantaneous aerosol DRE measured during the overpass. Figure 2a shows the evolution of the biomass burning season in 2006, with low DRE values in June, high values in July, extreme values in August and moderate values in September.

The area-averaged DRE of smoke over clouds reaches values up to 100 W m$^{-2}$ and more in mid-August 2006, according to SCIAMACHY and POLDER. The events during this period have been investigated often before (e.g. Chand et al., 2008; Jethva and Torres, 2011; Yu and Zhang, 2013). The SCIAMACHY DRE values were compared to model calculations from GCMs, particularly HadGEM2 (de Graaf et al., 2014). Models were not able to replicate these extremely high aerosol direct radiative effects. The emission of smoke from Africa was possibly strongly peaked in August, but even accounting for such episodic emissions in models did not explain the difference in aerosol effects in models and observations by SCIAMACHY. And Figure 2a shows that the aerosol DRE values from POLDER are even higher than those from SCIAMACHY. On the other hand, the average OMI-MODIS DRE is never higher than about 60 W m$^{-2}$.

The differences between the instruments are illustrated using histograms of all noon-normalized aerosol DREs, see Figure 3a. Clearly, the average POLDER aerosol DRE is almost twice as large as that from OMI-MODIS and SCIAMACHY (24.9 W m$^{-2}$ for OMI-MODIS, 28.4 W m$^{-2}$ for SCIAMACHY and 46.6 W m$^{-2}$ for POLDER). The statistics of the distributions are given in Table 3. In only the month August, the average aerosol DRE was 27.5 W m$^{-2}$ for OMI-MODIS, 36.8 W m$^{-2}$ for SCIAMACHY and 49.7 W m$^{-2}$ for POLDER. This is a somewhat larger difference between POLDER and SCIAMACHY than found by Peers et al. (2015) (about 10.5 W m$^{-2}$ difference between

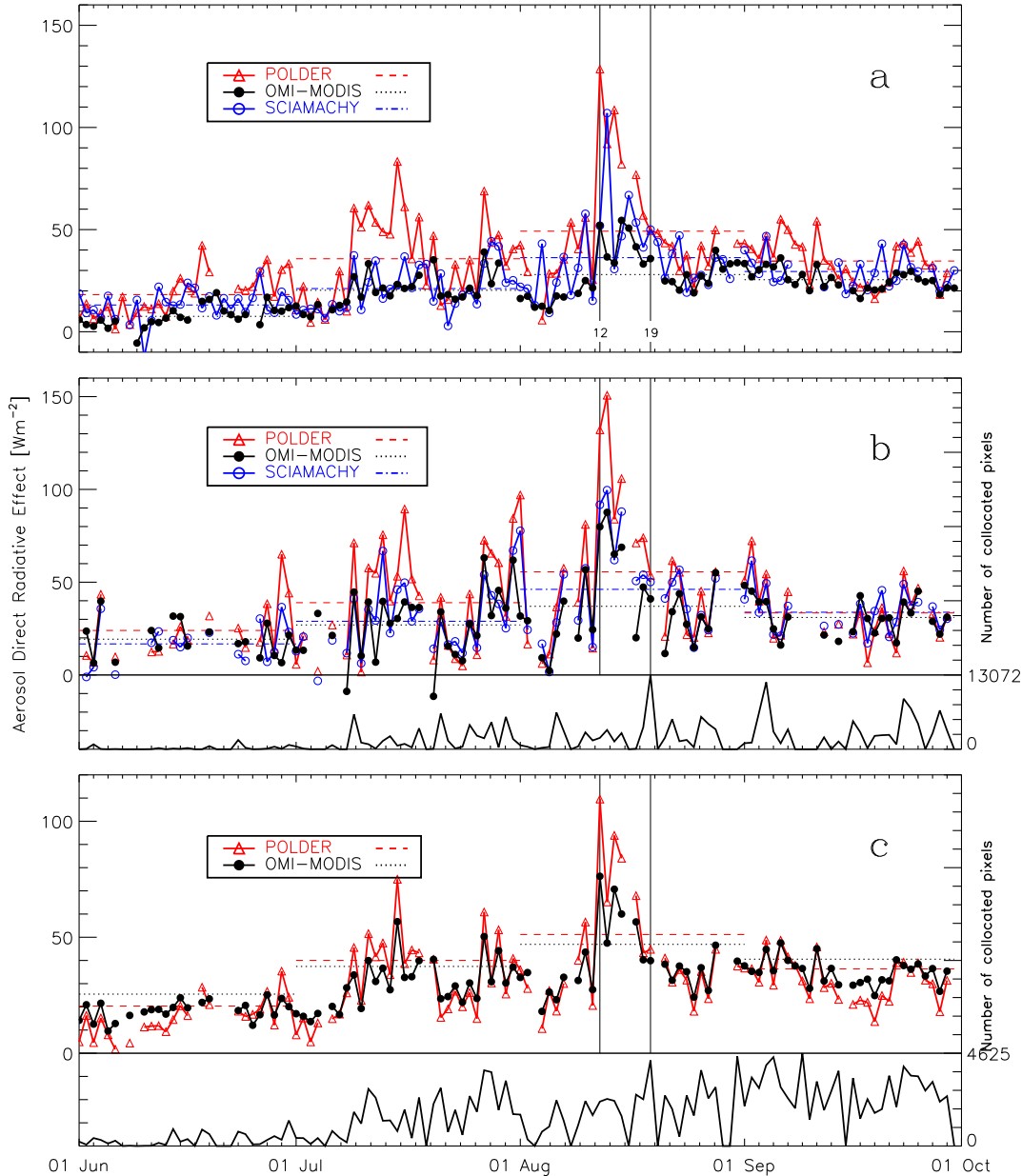

**Figure 2.** (a) Noon-normalized instantaneous aerosol DRE over clouds from combined OMI-MODIS reflectances (black), SCIAMACHY reflectances (blue) and POLDER AOT and COT retrievals (red) from 1 June - 1 October 2006, averaged over the area $10°$N–$20°$S;$10°$W–$20°$E in the south-east Atlantic. The average monthly aerosol DRE over clouds are given by the coloured straight lines during each month. (b) Same as (a), but for OMI-MODIS and SCIAMACHY pixels that were regridded to the $6 \times 6$ km$^2$ POLDER grid. Averaged values were only calculated from grid points that were covered by all three instruments. The number of collocated pixels that are covered by all three instruments is given in the lower panel in (b). (c) Area-averaged instantaneous aerosol DRE from OMI-MODIS and POLDER regridded to the OMI footprint. Note that because SCIAMACHY is omitted the number of pixels is much larger than in (a) and (b), and furthermore, the DRE is not noon-normalized, because the overpass time of OMI, POLDER and MODIS are similar.

SCIAMACHY and POLDER), but there POLDER DRE was averaged over a much larger area containing more small values of DRE.

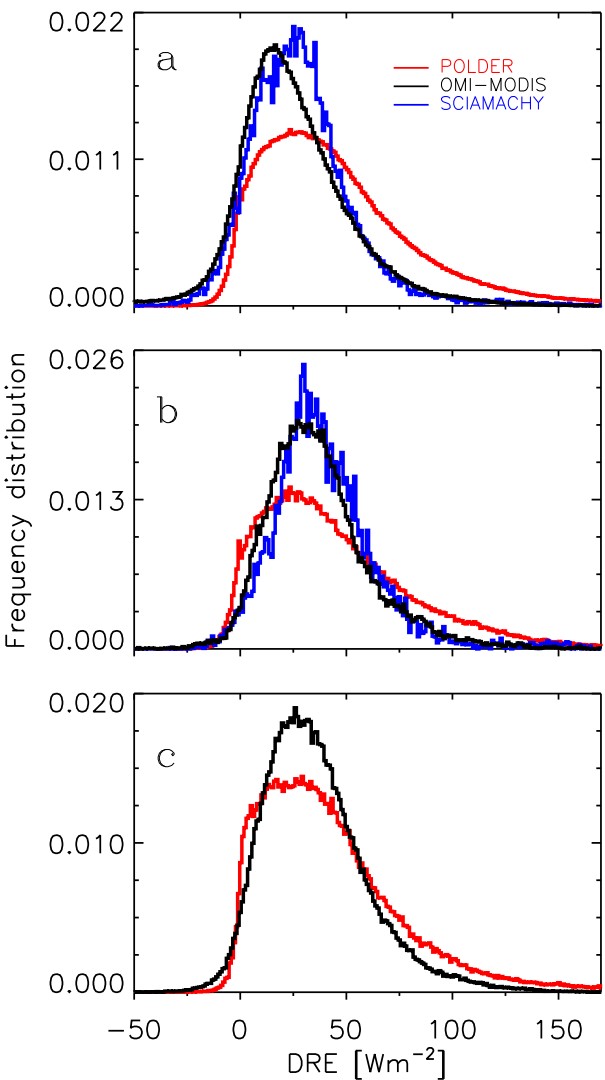

**Figure 3.** (a) Histograms of aerosol DRE over clouds in the Atlantic Ocean during June – September 2006 from POLDER AOT and COT retrievals (red), combined OMI-MODIS reflectance spectra (black) and SCIAMACHY reflectance spectra (blue). (b) Same as (a) but for collocated POLDER, OMI-MODIS regridded to POLDER grid and SCIAMACHY regridded to POLDER grid pixels only. (c) Same as (a) but for collocated POLDER regridded to OMI grid and OMI-MODIS pixels only.

The histograms show that the DRE from POLDER is higher than the DRE from OMI-MODIS mainly due to more high DRE values. This is indicated by the larger positive skewness for POLDER, a measure for the asymmetry of the distribution, where the other instruments show a more symmetric distribution.

**Table 3.** DRE Statistics of the different instruments before and after collocation for the area 10°N–20°S;10°W–20°E in the south-east Atlantic.

| Native grid | Mean | Median | Std. Dev | Skew |
|---|---|---|---|---|
| POLDER | 46.60 | 38.74 | 39.00 | 1.57 |
| OMI-MODIS | 24.88 | 21.92 | 30.27 | 0.62 |
| SCIAMACHY | 28.42 | 26.11 | 24.62 | 1.26 |
| Collocated POLDER grid | | | | |
| POLDER | 47.11 | 38.04 | 40.90 | 1.84 |
| OMI-MODIS | 37.13 | 33.74 | 26.15 | 1.65 |
| SCIAMACHY | 39.50 | 36.23 | 24.66 | 1.25 |
| Collocated OMI grid | | | | |
| POLDER | 43.66 | 36.30 | 35.91 | 1.62 |
| OMI-MODIS | 35.63 | 32.29 | 24.96 | 0.94 |

### 3.2.1 Sampling

Clearly, these different spatial scales limit the usefulness of a comparison of average values from satellite instruments. In order to correct for the issues described above, the OMI-MODIS and SCIAMACHY measurements were regridded onto a regular lat/lon grid, of 6666×3333 grid points. This corresponds to a 6 km × 6 km grid at the equator (reducing to 5.6×5.6 km$^2$ at 20°S). All regular grid cells covered by a SCIAMACHY or OMI pixel were given the value of that SCIAMACHY or OMI-MODIS DRE measurement. This gave SCIAMACHY and OMI-MODIS DRE values on a grid similar to the POLDER grid (albeit smoothed per OMI or SCIAMACHY pixel). The individual POLDER DRE values were then compared to the OMI-MODIS and SCIAMACHY DRE values in the grid cell that was closest to the POLDER grid cell. In Figure 2b the noon-normalized area-averaged instantaneous DRE over clouds over the south-east Atlantic is shown, like in Figure 2a, but using only those pixels that are covered by all three instruments. This effectively removes all sampling issues and differences due to different cloud screening strategies for the instruments. Note that at a number of days no values were available, since there were no areas with DRE that are sampled by all three instruments. This underlines the importance of sampling, even for such a fairly large area. The number of pixels over which was averaged per day is shown in the lower panel of Figure 2b.

The correlation between the noon-normalized area-averaged instantaneous DRE from the three instruments is now significantly improved compared to Figure 2a. The aerosol DRE from OMI-MODIS follows the aerosol DRE from SCIAMACHY very closely for almost the entire period shown. Note that the maximum DRE from OMI-MODIS is now increased to almost 90 W m$^{-2}$, which was due to removing many pixels with a moderate to low DRE during mid-August, that were not covered by POLDER and SCIAMACHY (cf. Figures 1a–c). Also note that the day with the largest average values does not occur on the 12$^{\text{th}}$ but on 13 August 2006 for all instruments, because SCIAMACHY

samples closer to the continent that day, where smoke plumes are generally thicker. The difference in average DRE between the instruments is also greatly reduced, see Figure 3b, which shows the histograms for only overlapping pixels re-
5 gridded to the POLDER grid, and its statistics in Table 3. The average DRE from POLDER is still about $47.0$ W m$^{-2}$ for only overlapping pixels, while the average DRE from OMI-MODIS has increased to $37.1$ W m$^{-2}$ and $39.5$ W m$^{-2}$ for SCIAMACHY regridded pixels.
Additionally, the sampling was checked by gridding the finer POLDER data to the coarser OMI grid and sampling only pixels that were covered by both OMI-MODIS and POLDER. In this case SCIAMACHY was omitted, so as not to lose too many POLDER and OMI-MODIS pixels because
of the poor SCIAMACHY sampling. The smaller POLDER pixels were averaged over the OMI footprint using a 2D Gaussian weighting function. This procedure is exactly the same for the averaging of MODIS pixels in an OMI footprint in the OMI-MODIS DRE computation, and described in de-
tail in de Graaf et al. (2016). Figures 2c and 3c show the area-averaged instantaneous aerosol DRE over clouds from col-located OMI-MODIS and POLDER pixels sampled on the OMI grid, and the histograms of both datasets. The statistics are given in Table 3. Obviously, gridding to the OMI grid
instead of to the POLDER grid does not change the results very much, but without SCIAMACHY the large number of pixels that are collocated results in a very high consistency between OMI-MODIS and POLDER DRE. Furthermore, without SCIAMACHY the noon-normalization is no longer
necessary because the overpass times of OMI, MODIS and POLDER are very close, and Figures 2c and 3c show the instantaneous local DRE as retrieved by the instruments, which helps the comparison discussion later on. The figures show that POLDER DRE is still larger than OMI-MODIS DRE,
especially for high values, but also lower for low values. This is illustrated by the skewness of the OMI-MODIS DRE distribution, which is now closer to the skewness of the distribution of POLDER DRE, but still the POLDER distribution is dominated by more high and low values.
This is also clear from a scatterplot of collocated POLDER DRE vs. OMI-MODIS DRE for regridded POLDER pixels, shown in Figure 4. The figure shows a good correlation between collocated POLDER and OMI-MODIS DRE, but with higher values for POLDER, especially for DRE larger
than $100$ W m$^{-2}$. An average ratio of OMI-MODIS DRE to POLDER DRE of $0.82$ can be found from Table 3, while a normal linear least-squares fit (shown by the red line in Figure 4) yields a slope of OMI-MODIS to POLDER ratio of only $0.63$. This is because the fit is dominated by the large
values, while the large majority of points are moderate values around $25$ W m$^{-2}$. When a fit is drawn which is weighted to the deviation from this moderate value (shown by the green line), a slope of $0.99$ is found, showing that the aerosol DRE over clouds is the same from POLDER and OMI-MODIS for
moderate values.

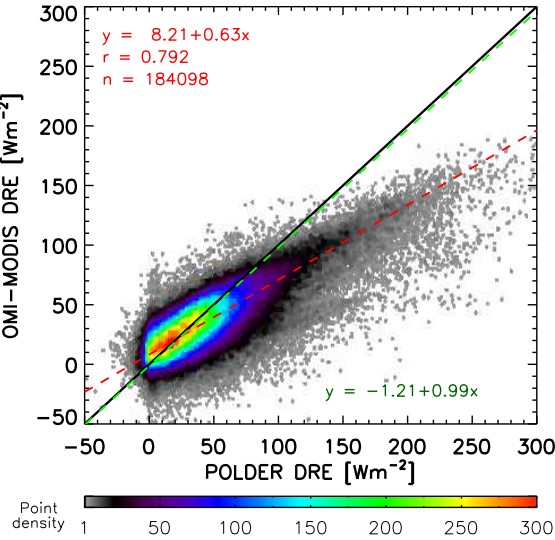

**Figure 4.** Scatterplot of POLDER DRE gridded to the OMI grid versus DRE from OMI-MODIS data from June-September 2006 over the south-east Atlantic. The red dashed line shows an unweighted linear least-squares fit. The green line shows the linear least-squares fit weighted by the distance to the average value of $25$ W m$^{-2}$.

## 3.3 Effects of differences in AOT and COT

To explain the difference between the OMI-MODIS DRE and the POLDER DRE, the differences in AOT and COT retrieved by the different instruments and their effects on the DRE are investigated. The effects on the DRE of a dif-
60 ference in AOT and COT are first investigated using simulated TOA spectra scenes of aerosols above clouds. An RTM was used to simulate the aerosol DRE at TOA of a scene with varying AOT, above a cloud deck with varying COT. For the simulations, a cloud was placed between 1 and 2
65 km and an aerosol layer between 2 and 5 km altitude. The clouds were simulated assuming a single-mode gamma particle size distribution with effective radius $r_{\mathrm{eff}} = 16\mu$m and an effective variance $\nu_{\mathrm{eff}} = 0.15$. For the aerosols, a bi-modal log-normal size distribution model was used, based on the
70 'very aged' biomass plume found over Ascension Island during SAFARI 2000 (Haywood et al., 2003). A refractive index of $1.54 - 0.018i$ was used for all wavelengths longer than $550$ nm. However, for the UV spectral region the imaginary refractive index was modified so that the absorption
Ångström exponent was $2.91$ in the UV, which fits satellite observations better (Jethva and Torres, 2011). The geometric radii for this haze plume used in the simulations were $r_{\mathrm{c}} = 0.255$ $\mu$m and $r_{\mathrm{f}} = 0.117$ $\mu$m for the coarse and fine modes, with standard deviations $\sigma_{\mathrm{c}} = 1.4$ and $\sigma_{\mathrm{f}} = 1.25$, re-
spectively. The fine mode number fraction was $0.9997$. These numbers are the same as used in de Graaf et al. (2012) to es-

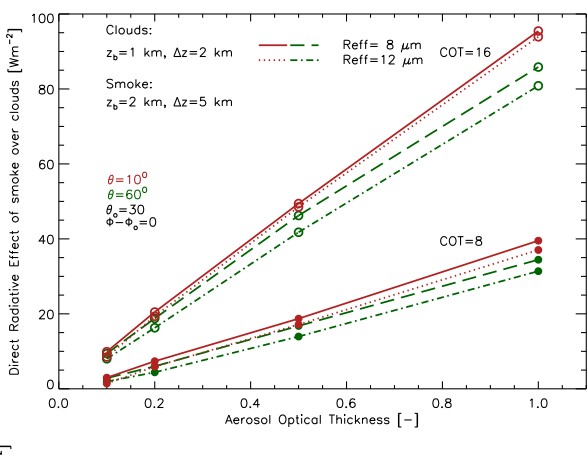

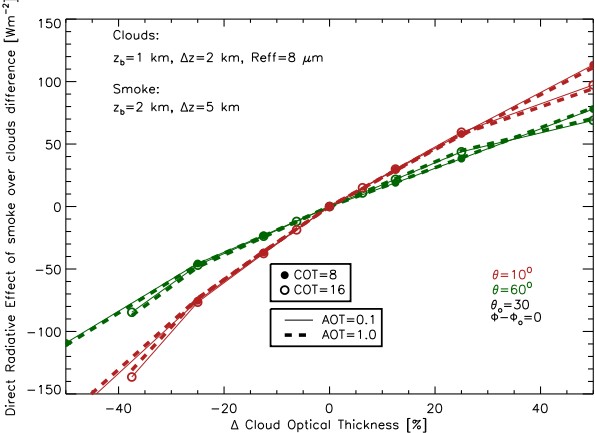

**Figure 5.** (a) Aerosol DRE for simulated scenes with clouds between 1-2 km and smoke aerosol between 2-5 km as a function of AOT at 550 nm. The COT was 8 or 16, the effective cloud droplet radius 8 or 12 $\mu$m. SZA was 30°, VZA was 10° or 60°, RAZI was 0°. (b) Aerosol DRE for simulated scenes as in (a), as a function of relative error in the retrieved COT.

timate the anisotropy change in the DRE calculation and in de Graaf et al. (2019) to study the BRDF of a scene with the aerosols above clouds.

The effects of varying AOT and varying COT on the aerosol DRE over clouds are illustrated in Figure 5a for an AOT between 0.1 and 1, and a COT of 8 and 16, with cloud effective radii of 8 and 12 $\mu$m. The solar zenith angle (SZA) in the simulations shown was 30°, the relative azimuth angle (RAZI) was 0° and two viewing zenith angles of 10 and 60° are shown, which span a typical range of viewing angles for OMI. The figure clearly shows a linear relationship between AOT and DRE, with an increasing aerosol DRE with increasing AOT, as expected. However, as known, the increase in DRE with AOT depends mainly on the COT of the underlying clouds. With larger COT, the amount of light at TOA increases, and the amount of absorption by the aerosols above

the clouds also increases, increasing the DRE. Clearly, the effect of AOT and COT on DRE are coupled. At a still relatively modest COT of 16, an increase of AOT from 0.1 to 1 increases the DRE from 10 to 95 W m$^{-2}$, for high AOT of 1, a doubling of COT from 8 to 16 increases the DRE from 40 to 95 W m$^{-2}$.

Accurate AOT and COT retrievals are clearly essential for an accurate aerosol DRE over clouds. For DAA, the effect of an error in the COT is estimated using simulated reflectances as above, shown in Figure 5b. An error of 20 % in COT can lead to an error in DRE of about 50 W m$^{-2}$, for COT in the range of 8–16, irrespective of the AOT. A note for DAA is in order here: The DRE is computed from the difference between a measured and simulated spectrum, which both have exactly the same COT and CER (since the simulation is done with the COT and CER retrieved from the measured spectrum). Therefore, any errors in the COT and CER retrieval have no influence on the difference between the two spectra and do not show in the DRE. However, if the COT or CER for the simulation were taken from a different measurement, however accurate, the simulated and measured spectra may be very different, giving rise to large DRE values, even without overlying aerosols. This was observed in a test where POLDER COT, regridded to the OMI grid, was used in the DRE computation, instead of the COT from the OMI-MODIS spectrum. Even though the POLDER COT was probably more accurate than the OMI-MODIS COT, the derived DRE was very erratic. For the POLDER DRE calculation this effect is different, because the DRE is computed using the scene twice with the same retrieved COT.

### 3.3.1 AOT differences

A part of the difference in DRE between OMI-MODIS (and SCIAMACHY) and POLDER can be attributed to differences in AOT, even if AOT is not explicitly retrieved for OMI-MODIS and SCIAMACHY. However, the AOT for small particles like smoke is assumed to be negligible in the SWIR from about 1.2 $\mu$m, which may be an underestimation. POLDER AOT, on the other hand, can be overestimated. Waquet et al. (2013a) estimated an overestimation of AOT for mineral dust above clouds of about 6 % due to plane parallel RTM computations. For smoke no estimate was given, but comparisons between AOT for smoke over clouds from several instruments show POLDER to be consistently on the high side, although not necessarily overestimated. On 12 august 2006, POLDER AOT at 550 nm was 1.1, averaged over the south-east Atlantic, with individual values up to 1.9. From CALIOP data an AOT of up to 1.5 (532 nm) was found for this day in the same area (Chand et al., 2009), while Jethva et al. (2013) found above-cloud AOT observed from MODIS up to 2.0. On 13 August 2006, the maximum OMI above cloud AOT was about 1.3, the maximum MODIS above cloud AOT about 1.5, the same as for POLDER (Jethva et al., 2014). Also from this day, POLDER above-

cloud AOT were about 11 % higher than the otherwise well correlated above-cloud AOT from the CALIOP depolarization ratio method (Deaconu et al., 2017). The effect of a high AOT can have a large effect on DRE for a high COT, according to Figure 5a. The average POLDER COT on 12 August 2006 was 12.9. A 6 % overestimation in AOT can lead to an overestimation in DRE of about 10 W m$^{-2}$ for that COT.

For SCIAMACHY and OMI-MODIS, on the other hand, the DRE is computed assuming negligible AOT at longer wavelengths, which is valid for sufficiently small particles. This assumption may break apart for larger particles, especially at very high AOT, leading to an underestimation of the AOT at higher values, limiting the DRE. The assumption can be tested by estimating the 1.2 $\mu$m AOT from POLDER AOT retrieved at 550 nm, using an Ångström exponent of 1.45, which was found in the spectral region from 325 to 1000 nm

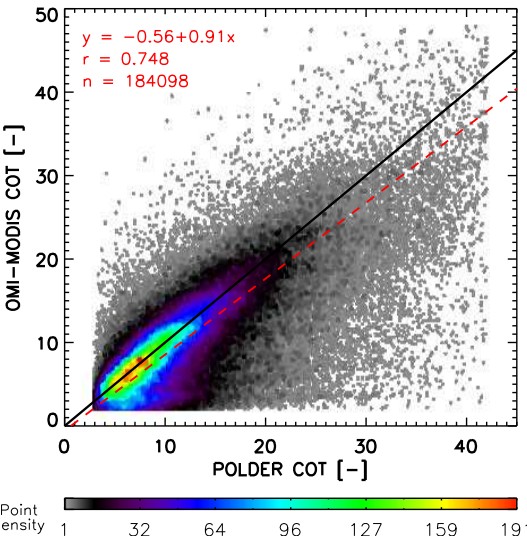

**Figure 7.** Scatterplot of POLDER COT gridded to the OMI footprint versus COT from OMI-MODIS data. The red dashed line shows the linear least-squares fit.

for African biomass burning aerosols from SAFARI 2000 observations (Bergstrom et al., 2007; Russell et al., 2010). This way, an AOT at 1.2 $\mu$m between 0.15 and 0.35 was found during the smoke peak in mid-August 2006, occasionally even reaching 0.6 (Schulte, 2016). The effect on the DRE was estimated at 21.7 W m$^{-1}\tau^{-1}$, by correcting the retrieved COT for the additional AOT (de Graaf et al., 2012), since this effect is essentially an underestimation of the COT. The effect is linear in AOT so the AOTs given by POLDER would result in an underestimation of the DRE by both SCIA-MACHY and OMI-MODIS of up to 13 W m$^{-2}$.

### 3.3.2 COT differences

The dependence of DRE on the COT of the cloud underlying the smoke can be large, depending on the AOT of the smoke. Figure 6 shows the COT retrievals from POLDER and OMI-MODIS on 12 August 2006, when the differences were the largest. POLDER COT is clearly higher peaked than OMI-MODIS COT. SCIAMACHY results are not shown, but similar to those from OMI-MODIS, although the COT may also be different because of the overpass by SCIAMACHY in the morning, when the cloud cover is systematically thicker than in the afternoon.

Note that POLDER COT is retrieved at 0.87 $\mu$m, while COT from OMI-MODIS is retrieved at 1.2 $\mu$m, which effectively is the MODIS measurement. However, the spectral variation in COT is very small and is only significant for very small droplets. For example, for cloud droplet effective radii of 4 microns the COT at 0.87 $\mu$m is about 4 % smaller than the COT at 1.2 $\mu$m and this reduces for larger droplets.

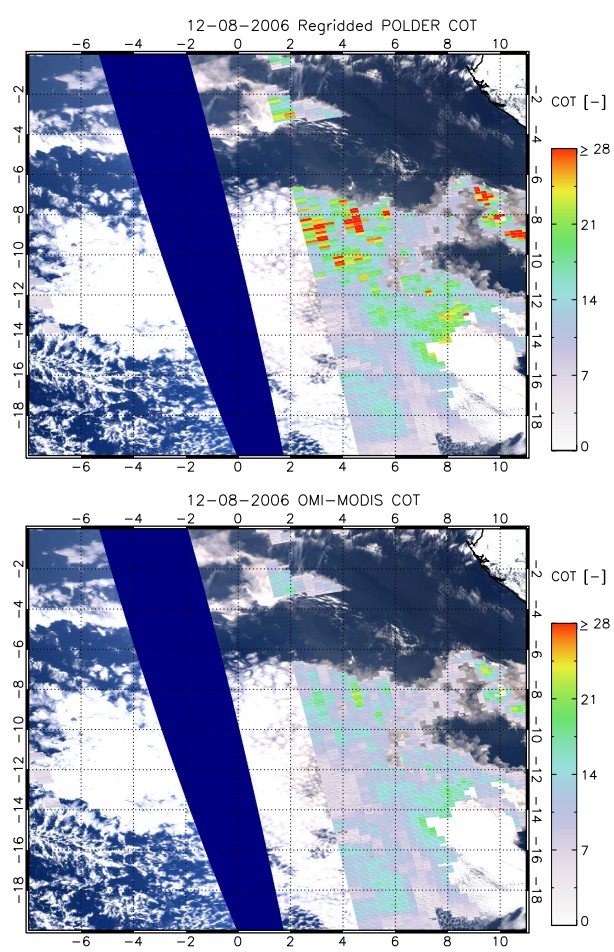

**Figure 6.** Cloud optical thickness (COT) overplotted on a MODIS RGB image on 12 August 2006 for POLDER at 0.87 $\mu$m regridded to OMI grid (upper panel) and OMI-MODIS at 1.2 $\mu$m (lower panel).

A comparison between MODIS and POLDER COT is presented in Figure 7 for all collocated OMI-MODIS and POLDER pixels regridded to the OMI footprint from June to September 2006. It shows that the COT from POLDER correlates well with the COT retrieved by MODIS but is higher by about 9 % on average. Here, POLDER COT values were averaged over the OMI footprint, while for MODIS the radiances were averaged and one COT is retrieved for that OMI pixel. Therefore, the difference in COT presented here can be caused by the plane parallel bias, which arises due to cloud heterogeneities and the nonlinear dependence of the cloud albedo (or reflectance) on water content (or optical thickness) (e.g. Oreopoulos and Davies, 1998). This effect is particular important in marine stratocumulus, which appear as plane parallel clouds, but are characterized by strong internal turbulent variability. Internal variability is largest at a cloud fraction of one and can have a stronger effect on the average (meso-scale) cloud optical thickness estimates than cloud fraction itself for marine stratocumulus clouds (e.g. Cahalan et al., 1994).

Again, the POLDER retrieval (of COT) is high, but not necessarily overestimated. An overestimation of 9 % at an average COT of about 13 and an AOT of 0.94 on 12 August 2006 would lead to a change of about 9 W m$^{-2}$. The average difference in COT is within the error estimate of the POLDER COT, but is more likely the result of the plane parallel bias.

Finally, the effect of different COT on the DRE retrieval was also computed using an RTM as used for the POLDER DRE calculations for the average values on the two selected days. The average COT from both POLDER and OMI-MODIS on 12 and 19 August 2006 for collocated pixels were determined and the above-cloud DRE was calculated for OMI-MODIS and POLDER using their mean COT and the mean AOT retrieved by POLDER. Results are summarized in Table 4. On 12 August 2006, the average COT from POLDER was 12.9, and from OMI-MODIS 10.0, while the average POLDER AOT was 0.94. This results in a DRE of 134 W m$^{-2}$ for POLDER and 107 W m$^{-2}$ for OMI-MODIS, while the average DRE on this day was 110 W m$^{-2}$ from POLDER and 76 W m$^{-2}$ from OMI-MODIS. On 19 August 2006, the mean COT from POLDER was 10.5, while from OMI-MODIS the mean COT was 10.5. Based on a mean AOT of 0.48, an aerosol DRE over clouds 63 W m$^{-2}$ and 53 W m$^{-2}$ was obtained from POLDER and OMI-MODIS, respectively, while the average values for that day were 45 and 40 W m$^{-2}$, respectively. Although the simulated average DRE is quite a bit larger than the average DRE found by the instruments, it suggests that the COT difference can account for about 80 % of the difference of 33 W m$^{-2}$ between the DRE from POLDER or OMI-MODIS shown in Figure 2.

**Table 4.** Average values of the OMI-MODIS COT and the POLDER COT and above-cloud AOT on 12 and 19 August 2006 between 20–0°S;8°W–14°E. The DRE was calculated using the average AOT from POLDER and the average COT values for both OMI-MODIS and POLDER, assuming a CER of 8 $\mu$m, an aerosol SSA of 0.840 at 550 nm and an aerosol geometric radius of 0.1$\mu$m.

| | Max COT | $\langle$COT$\rangle$ | POLDER $\langle$AOT$\rangle$ | simulated DRE | $\langle$DRE$\rangle$ |
|---|---|---|---|---|---|
| OMI GRID | | | | | |
| 12 August 2006 | | | | | |
| POLDER | 41.6 | 12.9 | 0.938 | 134.2 | 109.5 |
| OMI-MODIS | 37.5 | 10.0 | | 107.1 | 76.3 |
| 19 August 2006 | | | | | |
| POLDER | 41.6 | 10.5 | 0.477 | 63.0 | 44.8 |
| OMI-MODIS | 47.7 | 8.9 | | 53.1 | 39.9 |

## 4 Conclusions

In this paper, the aerosol direct radiative effect product is presented for cloud scenes in the south-east Atlantic, retrieved from SCIAMACHY reflectances, combined reflectance measurements from OMI and MODIS, and POLDER COT and AOT measurements in 2006. During this year, the production of smoke from vegetation fires in Africa was very large, and all instruments performed well. The average DRE from SCIAMACHY and OMI-MODIS, both retrieved using DAA, correlate very well, even though OMI-MODIS DRE has a much better resolution and coverage. The aerosol DRE from POLDER is completely independent. It correlates well with SCIAMACHY and OMI-MODIS DRE for moderate values, but is higher than SCIAMACHY and OMI-MODIS DRE for high values. The POLDER DRE is dependent on the retrieved AOT and COT, which in principle are both unbounded (although in the LUT for POLDER the COT is limited to 42). When the algorithm retrieves very large values for both, the derived DRE can also become very large. In mid-August, DRE above 300 W m$^{-2}$ were often reached, up to more than 400 W m$^{-2}$ for the noon-normalized DRE. This is 30 % of the maximum incoming solar irradiance. The DRE from SCIAMACHY and OMI-MODIS is limited to about 200–250 W m$^{-2}$ for individual pixels.

The largest contribution to the difference between SCIAMACHY, OMI-MODIS and POLDER DRE are sampling issues. Regridding SCIAMACHY and OMI-MODIS to the native POLDER grid and selecting only pixels sampled by all three instruments improved the comparison considerably. This approach removes issues related to filtering based on COT and CF, which can select high positive DRE values and lead to large differences in the average DRE. Even if the same filtering is used for the CF and COT for all instruments, different areas will be sampled, because the CF and COT retrieved by the different instruments may be different. After sampling, only smoothing due to the large footprints of SCIAMACHY and OMI remains, which is reflected in the

less extreme COT and DRE values compared to POLDER. This difference was reduced by gridding POLDER to the coarser OMI grid, improving the comparison between OMI-MODIS DRE and POLDER DRE. Because SCIAMACHY was not considered in this analysis, the statistics were much better than when SCIAMACHY collocation was required, because SCIAMACHY's spatial coverage is rather poor. The largest average difference after removing sampling issues between OMI-MODIS and POLDER DRE was 33 W m$^{-2}$ on 12 August 2006, which can be explained by different estimates of AOT and COT using the various instruments.

In DAA, the AOT is assumed to be zero at 1.2 $\mu$m, but was estimated from POLDER to be up to 0.6 in extreme cases, which results in an underestimation of the DRE in DAA of 13 W m$^{-2}$. For POLDER AOT, comparisons with OMI, MODIS and CALIOP AOT over clouds in the literature consistently show POLDER to be on the high side. POLDER AOT may be high-biased when aerosols are mixed into the cloud layer, enhancing the polarization signal. Also, when the smoke has a high real refractive index ($m_r > 1.47$) the AOT is overestimated by POLDER. However, the real part of the refractive index mostly impacts the scattering AOT (Peers et al., 2015), while the DRE calculation, in the case of biomass burning aerosols above clouds, is influenced mainly by the absorption AOT. The underestimation of the AOT for high values can explain about a third of the difference in DRE between POLDER and OMI-MODIS on 12 August 2006. POLDER AOT may be overestimated, but this is difficult to quantify.

The COT has a strong influence on the aerosol DRE over clouds. The average POLDER COT is about 9 % higher than that from OMI-MODIS in 2006. This difference can be caused by the plane parallel bias. Normally, MODIS COT retrievals at 0.8 and 2.1 $\mu$m retrievals are close to POLDER COT for fully clouded scenes with liquid water clouds (Zeng et al., 2012) (not considering overlying smoke). However, to avoid biases from smoke absorption, the MODIS channels at 1.2 and 2.1 $\mu$m are used to derive COT and CER for OMI-MODIS DRE, which may further influence the results. The difference between COT from OMI-MODIS and POLDER on 12 August 2006 can explain about 80 % of the difference in DRE on that day.

The errors in AOT and COT are not independent. In DAA, when the assumption of negligible AOT at longer wavelengths is no longer valid (large concentration of aerosols and/or large particles), the estimated COT is biased, resulting in a bias in the DRE. A better estimate of the DRE from the DAA method could be obtained when an unbiased retrieval of the COT was used, like e.g. from POLDER. However, a test of this approach using DDA on OMI-MODIS spectra but with POLDER COT yielded very erratic results. The reason is that the aerosol-free cloud spectrum simulated with POLDER COT can be quite different from the OMI-MODIS measured spectrum, yielding spectral differences that are interpreted as aerosol absorption. This may be improved if POLDER radiances were used.

This analysis shows that the aerosol direct effect of aerosols above clouds can be significant on the local scale when smoke is present over clouds. So far, model simulations have been unable to reproduce the high values, and many models underestimate the signal and even simulate a cooling (Zuidema et al., 2016), where the datasets in this analysis clearly show that the positive effect is significant and real. From the analysis here, we conclude that the aerosol DRE from OMI-MODIS and SCIAMACHY are likely underestimated, due to the bias in the cloud parameter retrieval when smoke is abundant. POLDER on the other hand, takes advantage of the polarization measurements to accurately estimate the COT, CER and AOT, without interdependent biases. However, for the spectral dependence of the aerosol absorption in the UV, there is still a dependence on the choice of aerosol model, and the aerosol DRE from POLDER may be on the high side. The two datasets from POLDER and OMI-MODIS most likely provide a high and low bound for the aerosol DRE in the south-east Atlantic, respectively, which can be used to challenge GCMs and test their aerosol intrinsic properties and aerosol-cloud-radiation interaction schemes. However, when observations and model simulations of local effects are compared, sampling issues should be properly accounted for, because area-averaging and time-averaging do not work well for episodic events like wildfire smoke plumes, which are short-lived and localized.

The analysis has shown the strengths and weaknesses of the DRE retrieval algorithms for POLDER, SCIAMACHY and OMI-MODIS. A combination of the two methods, DAA and DRE based on polarization measurements, could provide very accurate measurements of aerosol DRE over clouds, which is feasible for upcoming missions like METOP-SG-A and B (Marbach et al., 2015). These missions combine spectral imaging from a UV-VIS spectrometer Sentinel-5 and polarization measurements from a multi-angle polarimeter 3MI on one platform. The DAA method would benefit from unbiased COT retrievals, that could be provided with polarization measurements. The assumptions on the spectral dependence of the aerosol absorption in the POLDER-like retrieval can be assessed and improved by the DAA method in a closure study using the instruments on the METOP-SG platforms. This would allow time-dependent retrievals of UV-absorption by aerosols above clouds.

*Data availability.* The data used in this study are available from the authors. OMI-MODIS DRE are available online https://doi.org/10.21944/omi-modis-aerosol-direct-effect

*Author contributions.* MdG, LGT and PS are responsible for the DRE datasets from SCIAMACHY and OMI-MODIS. FP and FW

are responsible for the POLDER dataset. RS initially compared the datasets.

*Competing interests.* The authors declare no competing interests.

*Acknowledgements.* The work by MdG was funded by the Dutch National Programme for Space Research User of the Netherlands Space Office (NSO), project number ALW-GO/12-32.

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
