# Peer review of "Comparison of south-east Atlantic aerosol direct radiative effect over clouds from SCIAMACHY, POLDER and OMI-MODIS"

_Atmospheric Chemistry and Physics, 2019_

## Referee Comment (RC1) · Anonymous Referee #2 · 10 Sep 2019

This short study is a comparison of above-cloud aerosol direct radiative effects estimated by three methods applied to three satellite sensors or combinations of sensors (POLDER, SCIAMACHY, and OMI/MODIS). Looking at two days in August 2006 and at daily averages over 4 months in 2006, the authors find sizeable differences between the three sets of estimates, with POLDER retrievals producing significantly stronger radiative effects. Those differences are reduced when correcting for sampling differences. The remaining differences can be explained by differences in aerosol and cloud optical thickness, with cloud optical thickness being the dominant cause.

The study is of interest to the wider aerosol community because aerosol modellers

have now begun to use above-cloud aerosol retrievals to compare against their models, and large differences between observation-based estimates weaken observational constraints. This study is hopefully a first stage to eventually reconciling the different estimates. The paper is generally well-written, although language editing will help in places, and Figures and Tables illustrate the discussion well.

My main criticism of the study is that it does not attempt to bring additional information to resolve the disagreement. The discussion can also be improved in places. I recommend major revisions because addressing my main comment will probably require additional analyses.

**1 Main comment**

- The study concludes that differences are mostly caused, once the effect of sampling has been accounted for, by differences in cloud optical thickness (COT) retrievals between the instruments. Differences in aerosol optical thickness (AOT) also play a role, especially at longer wavelengths. But it would be most useful to know which dataset does best. Retrievals of AOT in nearby clear-sky regions, or using CALIOP, or even nearer the sources by AERONET should help determine whether the large AOTs (almost 2) retrieved by POLDER are realistic. Similarly, differences in retrieved COT are large enough to determine whether POLDER is realistic or not by comparing to CALIOP or passive retrievals, e.g. from SEVIRI. Adding such an analysis would make the study a more ambitious, and ultimately more useful, contribution.

**2 Other comments**

- Page 1, line 15: The statement "The effects of atmospheric aerosols are especially uncertain" repeats the first sentence and can be deleted.

- Page 1, line 21: I acknowledge that the terminology of aerosol direct, indirect, and semi-direct effects is now well known by the wider atmospheric science community, but I recommend defining them anyway for the sake of completeness.

- Page 2, line 2: "which can be characterized relatively well" sounds like an instance of concluding too quickly!

- Page 2, line 5: Caution: the use of "forcing" in the sense of Forster et al. 2007 implies that the unperturbed values correspond to pre-industrial conditions. In the present study however, unperturbed values are for an aerosol-free atmosphere, so to avoid confusion I recommend avoiding the word "forcing".

- Page 2, line 33: Myhre et al. (2013) is not the correct reference for that statement, as that paper only refers to global averages and does not isolate cloudy-sky radiative effects. I think the authors mean Figure 2 of Zuidema et al. (2016) doi:10.1175/BAMS-D-15-00082.1 . The same comment applies to Page 13, line 28.

- Page 3, line 15: "Finally, the . . . using an RTM." That has been said already.

- Page 3, line 16: "highest yet". What do you mean? Over which period are you making that statement?

- Page 5, line 4: "(from models)". Be more specific.

- Page 6, section 2.4: Isn't it possible to get an error/uncertainty for the POLDER product?

- Page 7, lines 3–4: How were the two cases selected?

- Page 7, section 3.2: That section is confusing. It goes back and forth between case studies and monthly averages. I suggest starting with case studies, then discussing the implications for longer time averages.

- Page 7, lines 30–31: "even of area-averages": I do not understand that statement.

- Page 9, section 3.2.1: The comparison protocol is unusual. The usual method is to regrid higher resolution datasets on to the coarser grids. The reason for doing like that is that the higher resolution represents variability within the coarser grid-box, so it is safe to make an average. But the authors do the other way around, replicating coarser values to fill the higher-resolution grid. Why that choice?

- Page 11, line 3: "it has been shown" requires a reference.

- Page 11, section 3.2.3: Why not show 12 Aug 2006 on Figure 5? The DRE difference is even larger on that day, which should help identify differences in COT as the main cause.

---

## Referee Comment (RC2) · Anonymous Referee #1 · 10 Sep 2019

**Review of deGraaf, et al., 2019.**

This paper examines satellite retrievals of the radiative effect of absorbing aerosols that overlie clouds (here termed the DRE). Retrievals from OMI+MODIS, POLDER and SCIAMACHY are compared. The latter can observe at many different wavelengths, but has low resolution. POLDER can observe the degree of polarization of the reflected light, which allows extra information about the aerosol and cloud to be obtained and minimizes the retrieval assumptions that need to be made. It is found that OMI+MODIS and SCIAMACHY agree reasonably well, but that POLDER produces larger DRE and cloud optical thicknesses (COT). Some of this difference is attributed to sampling issues (mainly arising from the different resolutions of the instruments) and some due to the larger optical depths retrieved by POLDER.

The study should be useful to other researchers since it would be useful to know how large this warming effect is (can it offset a significant amount of aerosol-cloud cooling?) and whether the models get it right. It also seems like the POLDER approach has some promise, particularly if it can be combined with more conventional instruments on e.g., the upcoming METOP-SG 3MI platform. As such I think it should be published after the suggested revisions.

However, the arguments are often a bit muddled and it would be good to see the reasons for the larger POLDER COT values explored a little more, as well as some more investigation into the effect of the low resolution retrievals from the other instruments. The paper talks a lot about "sampling errors" for OMI+MODIS and SCIAMACHY, but this seems to assume that all such errors are just from averaging of the final DRE or COT values, whereas it seems likely that some retrieval errors may be introduced by the averaging effects of the reflectances to low resolution, particularly if the relationship between the reflectances and the retrieved quantities are non-linear. Such effects occur for MODIS retrievals of effective radius and COD for example (Zhang; doi:10.1029/2012JD017655, 2012). It would be good to discuss this and to look into this possibility. It would even be possible to test what effects the averaging of reflectances to lower resolutions might have using synthetic higher resolution reflectances. On a similar note – considering just the "sampling effect" (i.e., just the effect of averaging the retrieved quantities, rather than the reflectances), it should be possible to quantify this effect by degrading the POLDER retrievals to the coarser grids, rather than the other way round, as is currently done.

Section 3.2.3 needs some checking as some of the statements regarding the POLDER optical depth being smaller seemed to contradict the results. The explanations were also not clear.

**Specific statements**

p.1 L15 – "Aerosol-cloud-radiation interactions currently present the largest uncertainty in our understanding of Earth's climate (Boucher et al., 2013). The effects of atmospheric aerosols are especially uncertain."

The second sentence here reiterates the first and does not really make sense. It should be removed, or else made more clear what it is referring to. Do you mean that the effects of aerosols alone are especially uncertain (compared to cloud-aerosol interactions)? However, I think that it is hard to argue that this case.

p. 1 L18 – "The presence of clouds has a strong influence on the DRE from the light absorbing species in smoke at TOA."

It's hard to understand what you mean here. I think you mean something like this :-

"The DRE (at TOA) due to the light absorbing species in smoke is strongly affected by the presence of clouds." Although maybe it would be good to introduce the idea of light absorption (rather than just scattering) affecting the DRE before this sentence. Or maybe this sentence isn't necessary given what follows?

p. 1 L20 – "Over clouds, on the other hand, scattering by aerosols is negligible" – this is not quite correct I think. The scattering due to aerosols overlying cloud would be quite high – it is the cloud that is doing less scattering in this case because of this. I think you mean that the addition of aerosols above a cloud has negligible extra impact on scattering relative to that which the cloud is already causing.

p.5, L2 – "CER was derived from collocated MODIS measurements."

Would it not be better for POLDER to retrieve the CER? Is this retrieval not possible? Could MODIS CER be biased by the overlying aerosol, or by inhomogeneous clouds, etc.?

p.5 L30 – "MODIS, on-board the Aqua satellite, flies in formation with Aqua in the A-Train, leading Aqua by about 15 minutes"

Should this be MODIS flies in formation with and leads Aura?

p.6 L14 - "Note however, that such an estimate is often missing, while methods other than DAA are moreover highly uncertain due to their dependence on the correct characterization of the spectral properties of the overlying aerosols."

This doesn't quite make sense. Do you mean that often such an error estimate is not made in other studies (does this only apply to those that use DAA)? Please correct if so. The part after should probably be a separate sentence.

p.6 L16 – "Other minor error sources for the DAA method are the uncertainty in input parameters, the influence of the smoke on the estimated cloud fraction, cloud optical thickness and cloud droplet effective radius, an uncertainty in the anisotropy factor (de Graaf et al., 2019), and the uncertainty of estimating the COT and CER at SWIR wavelengths."

In Section 3.2.3 you say that the DRE depends very strongly on the COT. So, wouldn't the COT uncertainty be likely to have a larger contribution to the error than indicated here? Also, this sentence needs to use semi-colons to make it clearer to become :-

"Other minor error sources for the DAA method are the uncertainty in input parameters; the influence of the smoke on the estimated cloud fraction, cloud optical thickness and cloud droplet effective radius; an uncertainty in the anisotropy factor (de Graaf et al., 2019); and the uncertainty of estimating the COT and CER at SWIR wavelengths."

p. 8, L2 – "the instantaneous aerosol DRE over clouds was normalized by dividing by the cosine of the solar zenith angle."

Have you checked whether the DREs scale linearly with the cosine of the angle (presumably a proxy for the incoming SW)? This could be checked with a radiative transfer code. If not then this might introduce some bias. Presumably there is a lower limit for the solar zenith angle allowed?

p. 8, L26 – "The main reason for the much larger area-averaged POLDER DRE on 12 August 2006 is the smaller coverage of the area by POLDER, compared to that by OMI/MODIS, due to a smaller swath. This was illustrated in Figures 1d and 1e."

This is part of the reason, but it seems that generally POLDER gives considerably higher values for the same regions. You could help demonstrate the magnitude of the differences caused by the different swath area vs that of POLDER values being higher by giving the collocated averages in Table 1. It would be better to change "The main reason" to "One of the reasons".

p. 8, L33 – "This means that a (dense) plume may be sampled once by a far off-center pixel, or by 15 nadir pixels, all of them receiving the same high values, depending on the satellite track."

For this to have an effect on the average it would require that the values retrieved from the mean reflectances over the larger pixel did not produce the correct average DRE value – i.e., there is a non-linear relationship between reflectance and the retrieved products, so that the result is dependent on the averaging scale (pixel size). It would be worth nothing this here. Also, the sentence would be clearer without "all of them receiving the same high values,".

p.10, L20 – "This issue could be resolved if all values were regridded to the coarsest available. However, since this is the SCIAMACHY grid, not many grid cells would remain."

Although you could do it for the OMI grid vs POLDER, which would be useful?

p.11 L19 – "A comparison of SCIAMACHY, OMI/MODIS and POLDER COT histograms (not shown) revealed a slightly higher COT from SCIAMACHY and OMI/MODIS compared to POLDER (up to 42 for POLDER and 48 for OMI/MODIS (Schulte, 2016)), but the maximum of POLDER is restricted due to LUT limits."

It's not clear here where or when these histograms apply to. I see that it is likely to refer to the 19[th] August case (Table 3), but it needs to be mentioned in the text. Also, "a slightly higher COT from SCIAMACHY" should be changed to "a slightly higher maximum COT from SCIAMACHY" since it otherwise it sounds like you are referring to mean values. However, visually it looks from Figure 5 like POLDER has higher maxima in general? You should also explain the part about the LUT limits in the context of the statement on p.10 L24 ("The POLDER DRE is dependent on the retrieved AOT and COT, which in principle are both unbounded.").

p. 11 L28 – "Even though the OMI/MODIS data are regridded to a high resolution grid, the values are obviously still more smoothed compared to the COT on the native high resolution POLDER grid. Therefore, even though POLDER COT and POLDER DRE are generally smaller than from OMI/MODIS on average, the extreme values and averages are higher."

The second sentence seems to contradict the rest of the paper – from the tables and figures POLDER has a generally larger DRE and COT?

Table 3 – it should be made clear in the table caption that the DRE was calculated using the POLDER AOT in both cases.

p.12 L8 – "It shows that the difference between these two quantities disappears completely for these instruments, and the slope is even reversed."

- It has reduced a lot, but not disappeared completely! Plus, saying that the slope has reversed is a bit unclear. Perhaps better to say it went from <1 to >1.

p. 12 L20 – "The aerosol DRE from POLDER is completely independent. It correlates well with SCIAMACHY and OMI/MODIS DRE for moderate values, but is larger than SCIAMACHY and OMI/MODIS DRE for high values. This is caused by a larger COT retrieved by POLDER, and to a lesser degree by an underestimation of the aerosol DRE using DAA, which by definition assumes a zero AOT at SWIR wavelengths.

The largest contribution to the difference between SCIAMACHY, OMI/MODIS and POLDER DRE are sampling issues."

   - It seems that the last sentence contradicts the ones before where it says that larger COT retrievals by POLDER are the cause. Is it the COT differences or the sampling issues that are most important? Or are they equally important? See also p.13 L8. Also, L9 in the abstract says that sampling issues are the most important – is this actually the case and can you point to the evidence that shows that it such errors are larger than the COT errors?

p.13 L5 – "This approach removes issues related to selecting high positive DRE values by

filtering on COT and CF, which introduce large differences in the average DRE."

- It's not clear what you are referring to here regarding filtering of COT and CF – is this a method that has been suggested in the literature (please say so and give a reference if so). Or from this paper – again this needs to me made clear.

p. 13 L13 – "Normally, MODIS COT retrievals at 0.8 and 1.2 microns retrievals"

   - Doesn't the usual MODIS retrieval over oceans use the 0.86 and 2.1um bands?

**Figures**
Fig. 2 – The linewidths of the monthly mean lines need to be quite a bit thicker for the colour and dash style to be visible.

Fig.3 – the legend lines need to be thicker to be able to see the different colours.

**Typos**
The word "microns" is used a lot, but also the symbol "µm". I think that the latter is the ACP standard for units.

p. 7, L20 – "when the comparison between the instrument is worst" -> "when the comparison between the instruments are worst"

p. 7, L30 – "Here, we show the effect of ignoring the sampling effect, even of area averages of, in this case, aerosol DRE over clouds."

   – this doesn't quite make sense. How about something more simple like "Here we show the effect of ignoring the sampling differences between instruments"?

p.10 L23 – "possibly" -> "possible".

p. 11 L12 – "This way, an AOT at 1.2 µm can be found between 0.15 and 0.35" -> "In this way an AOT at 1.2um of between 0.15 and 0.35 can be found"

p. 11, L22 – "However, the spectral variation in COT is very small. Only for very small cloud droplets the COT at 0.87microns is about 4% smaller than the COT at 1.2microns for cloud droplet effective radii of 4 microns, and this reduces for larger droplets."

- This would be better as :-

"However, the spectral variation in COT is very small and is only significant for very small droplets. For example, for cloud droplet effective radii of 4 microns the COT at 0.87microns is about 4% smaller than the COT at 1.2microns and this reduces for larger droplets."

p. 13 L23 – "Comparing AOT over clouds POLDER with MODIS and CALIOP, showed POLDER to be high, but not necessarily overestimated" – insert "from" between "clouds" and "POLDER".

p.13 L23 "om".

---

## Author Comment (AC1) · 28 Jan 2020

*Reviewer #1*
*This paper examines satellite retrievals of the radiative effect of absorbing aerosols that overlie clouds (here termed the DRE). Retrievals from OMI+MODIS, POLDER and SCIAMACHY are compared. The latter can observe at many different wavelengths, but*

[Figure]

*has low resolution. POLDER can observe the degree of polarization of the reflected light, which allows extra information about the aerosol and cloud to be obtained and minimizes the retrieval assumptions that need to be made. It is found that OMI+MODIS and SCIAMACHY agree reasonably well, but that POLDER produces larger DRE and cloud optical thicknesses (COT). Some of this difference is attributed to sampling issues (mainly arising from the different resolutions of the instruments) and some due to the larger optical depths retrieved by POLDER.*

*The study should be useful to other researchers since it would be useful to know how large this warming effect is (can it offset a significant amount of aerosol-cloud cooling?) and whether the models get it right. It also seems like the POLDER approach has some promise, particularly if it can be combined with more conventional instruments on e.g., the upcoming METOP-SG 3MI platform. As such I think it should be published after the suggested revisions.*

*However, the arguments are often a bit muddled and it would be good to see the reasons for the larger POLDER COT values explored a little more, as well as some more investigation into the effect of the low resolution retrievals from the other instruments. The paper talks a lot about 'sampling errors' for OMI+MODIS and SCIA-MACHY, but this seems to assume that all such errors are just from averaging of the final DRE or COT values, whereas it seems likely that some retrieval errors may be introduced by the averaging effects of the reflectances to low resolution, particularly if the relationship between the reflectances and the retrieved quantities are non-linear. Such effects occur for MODIS retrievals of effective radius and COD for example (Zhang; doi:10.1029/2012JD017655, 2012). It would be good to discuss this and to look into this possibility. It would even be possible to test what effects the averaging of reflectances to lower resolutions might have using synthetic higher resolution reflectances. On a similar note – considering just the 'sampling effect' (i.e., just the effect of averaging the retrieved quantities, rather than the reflectances), it should be possible to quantify this effect by degrading the POLDER retrievals to the coarser grids, rather*

*than the other way round, as is currently done.*

*Section 3.2.3 needs some checking as some of the statements regarding the POLDER optical depth being smaller seemed to contradict the results. The explanations were also not clear.*

The reviewer is thanked for the careful and thorough review of the manuscript. Many valuable suggestion were made, which were followed unless stated otherwise, in which a motivation is given. In particular, the regridding of POLDER data to the coarser grid of OMI was performed, to improve the comparison. This was not done the first time, because SCIAMACHY has the coarsest grid, and regridding to SCIAMACHY, and especially requiring SCIAMACHY collocation, yielded too sparse datasets. However, if only POLDER and OMI-MODIS are compared and collocated on the OMI grid, the analysis is much improved.

The manuscript was rewritten, to better distinguish between sampling issues and retrieval uncertainties. Sampling issues arise from the fact that different sensors in different orbits see different parts of the the atmosphere, and that different filter settings yield different pixels taken into account. This can be solved by requiring strict collocation of the considered pixels.

However, such collocation also requires resampling and regridding of data that are originally on different spatial resolutions. As the reviewer points out, nonlinear effects play a role here, and we have included a discussion on the role of the plane parallel bias for heterogeneous clouds. In our analyses MODIS radiances were added and resampled on the OMI footprint, while POLDER COT are averaged over the footprint. This has effects on the COT and CER averaging in a satellite footprint, and can account for the differences found between POLDER averaged COT and OMI-MODIS COT. We have tried to explain and quantify differences that we find. The resulting DRE differences are now explained in terms of the uncertainties in the AOT and COT retrievals. Additional

improvements of the measurements can then improve the DRE retrieval, but this is not the focus here.

All the issues raised by the reveiwer are addressed below:

**Specific statements**

*p.1 L15 Aerosol-cloud-radiation interactions currently present the largest uncertainty in our understanding of Earth's climate (Boucher et al., 2013). The effects of atmospheric aerosols are especially uncertain.*

*The second sentence here reiterates the first and does not really make sense. It should be removed, or else made more clear what it is referring to. Do you mean that the effects of aerosols alone are especially uncertain (compared to cloud-aerosol interactions)? However, I think that it is hard to argue that this case.*
Aerosol effect in global climate models are currently the largest uncertainty in global climate change attribution. However, the poor phrasing was also noted by reviewer #2, and the introduction was rewritten to better reflect the current state of aerosol climate science, and to clarify the text.

*p. 1 L18 – 'The presence of clouds has a strong influence on the DRE from the light absorbing species in smoke at TOA.*

*It's hard to understand what you mean here. I think you mean something like this : –*
*'The DRE (at TOA) due to the light absorbing species in smoke is strongly affected by the presence of clouds.' Although maybe it would be good to introduce the idea of light absorption (rather than just scattering) affecting the DRE before this sentence. Or maybe this sentence isn't necessary given what follows?*
I think an introductory sentence improves the paragraph, and the suggestion by the reveiwer was adopted as given.

*p. 1 L20 – 'Over clouds, on the other hand, scattering by aerosols is negligible' – this is not quite correct I think. The scattering due to aerosols overlying cloud would be quite high – it is the cloud that is doing less scattering in this case because of this. I think you mean that the addition of aerosols above a cloud has negligible extra impact on scattering relative to that which the cloud is already causing.*

Agreed. The text has been modified to read: 'Over clouds, on the other hand, scattering by aerosols hardly contribute to the upwelling radiation at TOA, since the scattering by clouds is dominant. However, the aerosols absorb radiation, lowering the planetary albedo, resulting in a positive direct effect (warming).'

*p.5, L2 – 'CER was derived from collocated MODIS measurements.' Would it not be better for POLDER to retrieve the CER? Is this retrieval not possible? Could MODIS CER be biased by the overlying aerosol, or by inhomogeneous clouds, etc.?*

POLDER does not have measurement in the near infrared. MODIS CER is retrieved primarily from the 2.1$\mu$m channel over the ocean. It can potentially be biased by the presence of aerosols above clouds. However, in the region of interest, the aerosols typically observed above clouds (i.e. biomass burning aerosols) are characterised by a large Ångström exponent. Therefore, their contribution to the signal at 2.1$\mu$m is expected to be negligible. This is the same argument that is used for the (OMI-)MODIS retrievals, except at 1.2$\mu$m. At 2.1$\mu$m the effect will be much smaller. Regarding the 3D effect, several filters are used on the POLDER AAC products in order to reject inhomogeneous clouds (Waquet *et al.*, 2013b, GRL)

*p.5 L30 – 'MODIS, on-board the Aqua satellite, flies in formation with Aqua in the A-Train, leading Aqua by about 15 minutes' Should this be MODIS flies in formation with and leads Aura?*

Correct, it should be (and is now): MODIS, on-board the Aqua satellite, flies in

formation with Aura in the A-Train, leading Aura by about 15 minutes.

*p.6 L14 - 'Note however, that such an estimate is often missing, while methods other than DAA are moreover highly uncertain due to their dependence on the correct characterization of the spectral properties of the overlying aerosols.'*

*This doesn't quite make sense. Do you mean that often such an error estimate is not made in other studies (does this only apply to those that use DAA)? Please correct if so. The part after should probably be a separate sentence.*

This is correct, an estimate on the individual measurements (of DRE in this case) is often missing. Many satellite products are delivered without uncertainty estimate on the individual measurements, e.g. relevant for this manuscript: OMI, MODIS, and CALIOP above cloud AOT. Uncertainty estimates are obtained from comparison with other datasets, like is done in this manuscript. However, we argue that error and uncertainty estimates can, and should, also be given on the basis of assumptions and uncertainties of the input parameters, which lead to measurement uncertainties. In that case, comparisons like the current one, can be performed in light of the uncertaities of the measurements.

Here, we have tried to quantify the uncertainties in aerosol DRE in terms of uncertainty estimates in above cloud AOT and COT for POLDER, and relate the difference between OMI-MODIS and POLDER DRE in terms of those uncertainties.

*p.6 L16 – 'Other minor error sources for the DAA method are the uncertainty in input parameters, the influence of the smoke on the estimated cloud fraction, cloud optical thickness and cloud droplet effective radius, an uncertainty in the anisotropy factor (de Graaf et al., 2019), and the uncertainty of estimating the COT and CER at SWIR wavelengths.'*

*In Section 3.2.3 you say that the DRE depends very strongly on the COT. So, wouldn't*

*the COT uncertainty be likely to have a larger contribution to the error than indicated here? Also, this sentence needs to use semi-colons to make it clearer to become :-*

*'Other minor error sources for the DAA method are the uncertainty in input parameters; the influence of the smoke on the estimated cloud fraction, cloud optical thickness and cloud droplet effective radius; an uncertainty in the anisotropy factor (de Graaf et al., 2019); and the uncertainty of estimating the COT and CER at SWIR wavelengths.'*
Yes and no. The (large) effect of the COT uncertainty on DRE is in the uncertainty estimate of the DRE by applying it to aerosol-free cloud scenes, which yields the rather large uncertainty of 13 Wm$^{-2}$ for OMI-MODIS DRE. The additional cloud uncertainties investigated in De Graaf *et al.* (2012) are the effect *of smoke* on the cloud parameters, and uncertainty of estimating CER and COT in the SWIR *instead of in the visible*. However, it is agreed that this is not clear from the text. Futhermore, we show in this paper that the effects of COT and AOT are coupled and COT uncertainties will have larger effects at larger above clouds AOT, which was not estimated. This was added to the manuscript.

*p. 8, L2 – 'the instantaneous aerosol DRE over clouds was normalized by dividing by the cosine of the solar zenith angle.'*

*Have you checked whether the DREs scale linearly with the cosine of the angle (presumably a proxy for the incoming SW)? This could be checked with a radiative transfer code. If not then this might introduce some bias. Presumably there is a lower limit for the solar zenith angle allowed?*

The DRE is defined as the difference in upwelling flux at TOA for a cloud scene and a cloud with aerosol scene, which can be written as

$$\text{DRE}_{\text{aer}} = \mathsf{F}^{\uparrow}_{\text{cld+aer}} - \mathsf{F}^{\uparrow}_{\text{cld}} = \mu_0 \int_{SW} E_0(\lambda)(A_{\text{cld}} - A_{\text{cld+aer}}) \, \mathrm{d}\lambda \tag{1}$$

where $A$ is the local planetary albedo, defined in terms of the reflectance $R = \pi I / \mu_0 E_0$ as $A(\mu_0, \lambda) = \frac{1}{\pi} \int_0^{2\pi} \int_0^1 R(\lambda; \mu, \phi; \mu_0, \phi_0) \mu d\mu_0 d\phi$. Neglecting the small effect on the planetary albedo (which is the effect of the BRDF of the cloud scenes, that has been treated in De Graaf *et al.* (2012) and De Graaf *et al.* (2019)), the fluxes can be noon-normalised by deviding DRE by $\mu_0$ to cancel the effect of different solar incoming fluxes $\mu_0 E_0$ on the DRE computation during the overpasses of SCIAMACHY (10:00 LT) and OMI, MODIS and POLDER (around 13:30 LT).

The solar zenith angle has no lower limit, since this does not produce any problems. High solar zenith angles may introduce biases, but these do not occur, because the considered area of the south-east Atlantic basin is near the equator.

*p. 8, L26 – 'The main reason for the much larger area-averaged POLDER DRE on 12 August 2006 is the smaller coverage of the area by POLDER, compared to that by OMI/MODIS, due to a smaller swath. This was illustrated in Figures 1d and 1e.'*

*This is part of the reason, but it seems that generally POLDER gives considerably higher values for the same regions. You could help demonstrate the magnitude of the differences caused by the different swath area vs that of POLDER values being higher by giving the collocated averages in Table 1. It would be better to change 'The main reason' to 'One of the reasons'.*

All the sampling issues are resolved when the OMI-MODIS data are regridded to POLDER or vv, which is shown in the manuscript. The description here is merely to illustrate the main difference between the two datasets on 12 August 2006, which is not the day with the most extreme DRE values, but the day with the largest difference between the area-averages. The reason is the sampling, which is very different for

OMI-MODIS and POLDER, as shown in Figure 1d–e. When the sampling issue is removed the average for that day reduced from 74 Wm$^{-2}$ difference to 33 Wm$^{-2}$ difference. So the sampling issue is the main reason for the large descrepancy. The numbers are added to table 3.

*p. 8, L33 – 'This means that a (dense) plume may be sampled once by a far off-center pixel, or by 15 nadir pixels, all of them receiving the same high values, depending on the satellite track.'*

*For this to have an effect on the average it would require that the values retrieved from the mean reflectances over the larger pixel did not produce the correct average DRE value – i.e., there is a non-linear relationship between reflectance and the retrieved products, so that the result is dependent on the averaging scale (pixel size). It would be worth nothing this here. Also, the sentence would be clearer without 'all of them receiving the same high values,'.*

This unclear statement has been removed. The text has been changed to just state the different pixel sizes. The discussion has been changed to showing the effect of regridding, both SCIAMACHY and OMI-MODIS to the high-resolution POLDER grid, as POLDER to the OMI grid.

*p.10, L20 – 'This issue could be resolved if all values were regridded to the coarsest available. However, since this is the SCIAMACHY grid, not many grid cells would remain.'*

*Although you could do it for the OMI grid vs POLDER, which would be useful?*

Yes. The main addition to the new manuscript is the regridding of POLDER DRE to the coarser OMI grid. The text was rewritten to include this analysis. Without the SCIAMACHY collocation requirement the coverages of both datasets are still very good after collocation and sampling issues are removed. It shows that regridding to

POLDER grid or OMI grid does not change the results very much, only the removal of the SCIAMACHY collocation requirement ensures much better statistics.

*p.11 L19 – 'A comparison of SCIAMACHY, OMI/MODIS and POLDER COT histograms (not shown) revealed a slightly higher COT from SCIAMACHY and OMI/MODIS compared to POLDER (up to 42 for POLDER and 48 for OMI/MODIS (Schulte, 2016)), but the maximum of POLDER is restricted due to LUT limits.'*

*It's not clear here where or when these histograms apply to. I see that it is likely to refer to the 19 August case (Table 3), but it needs to be mentioned in the text. Also, 'a slightly higher COT from SCIAMACHY' should be changed to 'a slightly higher maximum COT from SCIAMACHY' since it otherwise it sounds like you are referring to mean values. However, visually it looks from Figure 5 like POLDER has higher maxima in general? You should also explain the part about the LUT limits in the context of the statement on p.10 L24 ('The POLDER DRE is dependent on the retrieved AOT and COT, which in principle are both unbounded.').*

The discussion on COT was completely rewritten and this text was removed. We believe the COT analysis is now much clearer, the relationship between POLDER COT and OMI-MODIS COT is now shown in Figure 6, with POLDER COT being systematically higher.

The statement on p.10 L24 is rephrased as: The POLDER DRE is dependent on the retrieved AOT and COT, which in principle are both unbounded (although in the LUT for POLDER THE COT is limited to 42).

*p. 11 L28 – 'Even though the OMI/MODIS data are regridded to a high resolution grid, the values are obviously still more smoothed compared to the COT on the native high resolution POLDER grid. Therefore, even though POLDER COT and POLDER DRE are generally smaller than from OMI/MODIS on average, the extreme values and*

*averages are higher.'*

*The second sentence seems to contradict the rest of the paper – from the tables and figures POLDER has a generally larger DRE and COT?*
The discussion on COT was completely rewritten and this text was removed.

*Table 3 – it should be made clear in the table caption that the DRE was calculated using the POLDER AOT in both cases.*
Agreed, this was added to the caption, and the value itself was placed in the center, so it is obvious it was not retrieved for OMI-MODIS.

*p.12 L8 – 'It shows that the difference between these two quantities disappears completely for these instruments, and the slope is even reversed.'*

*- It has reduced a lot, but not disappeared completely! Plus, saying that the slope has reversed is a bit unclear. Perhaps better to say it went from <1 to >1.*
The discussion on COT was completely rewritten. Only OMI-MODIS and POLDER data were used to examine the DRE disfferences. The relationship between POLDER COT and OMI-MODIS COT is now shown in Figure 6.

*p. 12 L20 – 'The aerosol DRE from POLDER is completely independent. It correlates well with SCIAMACHY and OMI/MODIS DRE for moderate values, but is larger than SCIAMACHY and OMI/MODIS DRE for high values. This is caused by a larger COT retrieved by POLDER, and to a lesser degree by an underestimation of the aerosol DRE using DAA, which by definition assumes a zero AOT at SWIR wavelengths.*

*The largest contribution to the difference between SCIAMACHY, OMI/MODIS and POLDER DRE are sampling issues.'*

*- It seems that the last sentence contradicts the ones before where it says that larger*

*COT retrievals by POLDER are the cause. Is it the COT differences or the sampling issues that are most important? Or are they equally important? See also p.13 L8. Also, L9 in the abstract says that sampling issues are the most important – is this actually the case and can you point to the evidence that shows that it such errors are larger than the COT errors?*

Sampling differences are the main source for DRE differences, which is now clearer from the case of 12 August 2006, see above. Only after sampling is removed an analysis of the differences in terms of the input parameters and assumptions makes sense. After removing sampling issues, COT differences are the largest contribution to the found differences. AOT differences have a smaller but still important effect, and AOT and COT influences are coupled. The new manuscripts makes these conclusions more apparent, thanks to the suggestions of the reviewers.

*p.13 L5 – 'This approach removes issues related to selecting high positive DRE values by filtering on COT and CF, which introduce large differences in the average DRE.'*

*It's not clear what you are referring to here regarding filtering of COT and CF – is this a method that has been suggested in the literature (please say so and give a reference if so). Or from this paper – again this needs to me made clear.*

The sampling by the three instruments is different. Not only because they sense different areas, but also because even if the same limits on permitted cloud fractions and COT are used, the results will be different, simply because the retrieved CFs and COTs may be different. Therefore, even perfectly aligned instruments with exactly the same filter settings will sample different parts of the Earth. These issues are also resolved with the collocation requirement used in the analysis.

A sentence explaining this was added to the manuscript: 'Even if the same filtering is used for the CF and COT for all instruments, different areas will be sampled, because the CF and COT retrieved by the different instruments may be different.'

*p. 13 L13 – 'Normally, MODIS COT retrievals at 0.8 and 1.2 microns retrievals' - Doesn't the usual MODIS retrieval over oceans use the 0.86 and 2.1um bands?*
Correct, this has been changed to 2.1 microns.

**Figures** *Fig. 2 – The linewidths of the monthly mean lines need to be quite a bit thicker for the colour and dash style to be visible.*
Agreed.

*Fig.3 – the legend lines need to be thicker to be able to see the different colours.*
Agreed

**Typos**

*The word 'microns' is used a lot, but also the symbol 'μm'. I think that the latter is the ACP standard for units.*
Agreed. All are changed to $\mu$m.

*p. 7, L20 – 'when the comparison between the instrument is worst' -> 'when the comparison between the instruments are worst'*
'instrument' was changed to 'instruments'. (the comparison ..) 'is' was retained.

*p. 7, L30 – 'Here, we show the effect of ignoring the sampling effect, even of area averages of, in this case, aerosol DRE over clouds.'*
The text was changed.

*– this doesn't quite make sense. How about something more simple like 'Here we show the effect of ignoring the sampling differences between instruments'?*

Agreed. I also don't understand this sentence. The reviewer's suggestion was adopted.

*p.10 L23 – 'possibly' -> 'possible'.*
Correct. Text was changed.

*p. 11 L12 – 'This way, an AOT at 1.2μm can be found between 0.15 and 0.35' -> 'In this way an AOT at 1.2um of between 0.15 and 0.35 can be found'*
Agreed.

*p. 11, L22 – 'However, the spectral variation in COT is very small. Only for very small cloud droplets the COT at 0.87microns is about 4% smaller than the COT at 1.2microns for cloud droplet effective radii of 4 microns, and this reduces for larger droplets.'*

*- This would be better as :- 'However, the spectral variation in COT is very small and is only significant for very small droplets. For example, for cloud droplet effective radii of 4 microns the COT at 0.87microns is about 4% smaller than the COT at 1.2microns and this reduces for larger droplets.'*
Gladly accepted.

*p. 13 L23 – 'Comparing AOT over clouds POLDER with MODIS and CALIOP, showed POLDER to be high, but not necessarily overestimated' – insert 'from' between 'clouds' and 'POLDER'.*
Agreed. Text was changed.

*Reviewer #2*
*This short study is a comparison of above-cloud aerosol direct radiative effects estimated by three methods applied to three satellite sensors or combinations of sensors (POLDER, SCIAMACHY, and OMI/MODIS). Looking at two days in August 2006 and at daily averages over 4 months in 2006, the authors find sizeable differences between the three sets of estimates, with POLDER retrievals producing significantly stronger radiative effects. Those differences are reduced when correcting for sampling differences. The remaining differences can be explained by differences in aerosol and cloud optical thickness, with cloud optical thickness being the dominant cause. The study is of interest to the wider aerosol community because aerosol modellers have now begun to use above-cloud aerosol retrievals to compare against their models, and large differences between observation-based estimates weaken observational constraints. This study is hopefully a first stage to eventually reconciling the different estimates. The paper is generally well-written, although language editing will help in places, and Figures and Tables illustrate the discussion well. My main criticism of the study is that it does not attempt to bring additional information to resolve the disagreement. The discussion can also be improved in places. I recommend major revisions because addressing my main comment will probably require additional analyses.*

**1 Main comment** *The study concludes that differences are mostly caused, once the effect of sampling has been accounted for, by differences in cloud optical thickness (COT) retrievals between the instruments. Differences in aerosol optical thickness (AOT) also play a role, especially at longer wavelengths. But it would be most useful to know which dataset does best. Retrievals of AOT in nearby clear-sky regions, or using CALIOP, or even nearer the sources by AERONET should help determine whether*

*the large AOTs (almost 2) retrieved by POLDER are realistic. Similarly, differences in retrieved COT are large enough to determine whether POLDER is realistic or not by comparing to CALIOP or passive retrievals, e.g. from SEVIRI. Adding such an analysis would make the study a more ambitious, and ultimately more useful, contribution.*

The reviewer is thanked for the careful evaluation of the manuscript. We have changed many of the discussions, which were indeed sometimes vague, because the results were not always clear. After reanalysis of the sampling issues, by gridding POLDER to OMI and disregarding SCIAMACHY to get a good comparison between two datastes with a good coverage, the results are much clearer, which are -hopefully- clearer presented in the new manuscript.

After sampling, the remaining differences between POLDER and OMI-MODIS DRE are explained in terms of AOT and COT and the uncertainties in those. The uncertainties in COT retrieved by POLDER and OMI-MODIS for polluted clouds are difficult to establish, because there little to compare with, 'normal' COT retrievals being biased by overlying smoke. Both POLDER COT and OMI-MODIS COT retrievals show continuous behaviour from polluted to unpolluted areas. We show the difference in COT retrieval (9% on average, no extreme differences) and the effect on the DRE.

For AOT, the comparison with more established datasets is also difficult, because these are all in clear sky. However, an inspection of several AERONET sites showed high AOT during the biomass burning seasons, but never as high as 2 over the Atlantic. Ascension Island (almost 3000 km from the source) has no measurements during 2006, but other years AOT is measured up to 1 (UV). e.g. in 2016, which was also an anomalously extensive biomass burning season. St. Helena has few measurements, São Tomé has measurements in 2017 and 2018 up to 1.5 (UV). Only over Gabon, which is most likely the source region or in the path of the smoke towards the ocean, AOT at 340 and 280 nm of more than 2.0 was found in August 2016 (data start in 2014).

We have compared the above cloud AOT (ACAOT) from several sources that are currently available, from MODIS, OMI, and CALIOP, all of them science datasets, i.e. no proper validation has been performed for these datasets. High ACAOT up to 1.5 or 2 is common, with POLDER being in line with the highest retrievals. The discussion in the manuscript was extended with these numbers from the literature.

The DRE results and differences are explained in terms of these findings, but no conclusion was given on which dataset is 'best', because the truth is not known.

**2 Other comments** *Page 1, line 15: The statement 'The effects of atmospheric aerosols are especially uncertain' repeats the first sentence and can be deleted.*
Agreed. The introduction was rewritten to be more clear and correct.

*Page 1, line 21: I acknowledge that the terminology of aerosol direct, indirect, and semi-direct effects is now well known by the wider atmospheric science com- munity, but I recommend defining them anyway for the sake of completeness.*
Agreed. The introduction was rewritten to include this.

*Page 2, line 2: 'which can be characterized relatively well' sounds like an in- stance of concluding too quickly!*
We derive a direct effect of aerosols over clouds including an uncertainty estimate. I think this a relatively good performance, given that semi-direct effetcs are not even estimated at all from measurements.

*Page 2, line 5: Caution: the use of 'forcing' in the sense of Forster et al. 2007 implies that the unperturbed values correspond to pre-industrial conditions. In the present study however, unperturbed values are for an aerosol-free atmosphere, so to avoid confusion I recommend avoiding the word "forcing".*

In 2007, Forster *et al.* defined 'radiative forcing' as the net broadband irradiance change ΔF at a certain level with and without the forcing constituent, after allowing for stratospheric temperatures to readjust to radiative equilibrium, but with tropospheric and surface temperatures and state held fixed at the unperturbed values (chapter 2.2.). This was quoted in the manuscript, with 'radiative forcing' changed to 'radiative effect', because the terminology has changed since then in favour of 'forcing' as the change since per-industrial times, and 'effect' as the instantaneous change. However, one instance of 'radiative forcing' on p2,l7 was overlooked, and this was changed to 'radiative effect' in the new manuscript.

*Page 2, line 33: Myhre et al. (2013) is not the correct reference for that statement, as that paper only refers to global averages and does not isolate cloudy- sky radiative effects. I think the authors mean Figure 2 of Zuidema et al. (2016) doi:10.1175/BAMS-D-15-00082.1 . The same comment applies to Page 13, line 28.*
Agreed.

*Page 3, line 15: 'Finally, the . . . using an RTM.' That has been said already.*
Agreed.

*Page 3, line 16: 'highest yet'. What do you mean? Over which period are you making that statement?*
Over the south-east Atlantic in 2006, as stated in the manuscript.

*Page 5, line 4: '(from models)'. Be more specific.*
This has been removed.

*Page 6, section 2.4: Isn't it possible to get an error/uncertainty for the POLDER prod-*

*uct?*

The main source of error for the aerosol DRE over clouds from POLDER is the assumption on the aerosol refractive index. In the first step of the algorithm, an assumption on the refractive index is used in order to retrieve the above-cloud scattering AOT. In the second step, the imaginary part is modified in order to retrieve the absorption AOT from total reflectances, assuming the same real part of the refractive index as in the first step. The impact of the refractive index assumption on the DRE has been analysed in Peers et al (2015) and a maximum error of 10 Wm$^{-2}$ has been observed. Finally, an error on the CER can cause a bias of up to 10% on the COT.

This was added to the manuscript.

*Page 7, lines 3–4: How were the two cases selected?*

The first case shows the situation during the largest difference between the datasets, and the second case the situation one week later, when the differences are small. During 2006 all instruments performed well, and August is the peak of the biomass burning season in southern Africa.

This was added to the manuscript.

*Page 7, section 3.2: That section is confusing. It goes back and forth between case studies and monthly averages. I suggest starting with case studies, then discussing the implications for longer time averages.*

The discussion now starts with the cases, and the discussion on the aera-averaged DRE goes back to the cases, which are part of the dataset. In Figure 2, the 12th and 19th are indicated more clearly, so the reader understands that Figure 1 and 2 are connected, and where.

*Page 7, lines 30–31: 'even of area-averages': I do not understand that statement.*

[Figure]

As stated, sampling effects are often treated by averaging. Here, we show that this is not sufficient for the sparse DRE over the Atlantic. The text 'even of area-averages' was removed.

*Page 9, section 3.2.1: The comparison protocol is unusual. The usual method is to regrid higher resolution datasets on to the coarser grids. The reason for doing like that is that the higher resolution represents variability within the coarser grid- box, so it is safe to make an average. But the authors do the other way around, replicating coarser values to fill the higher-resolution grid. Why that choice?*
The reason was to avoid the very coarse SCIAMACHY grid. However, another analysis was added without SCIAMACHY and with POLDER gridded to OMI, which is the 'normal' way, and still has a large coverage from both instruments. This improved the comparison considerably. Both the reason for the first choice, and the new comparison were added to the manuscript.

*Page 11, line 3: 'it has been shown' requires a reference.*
The statement was from the reference just before the sentence. The reference was moved to include this statement as well.

*Page 11, section 3.2.3: Why not show 12 Aug 2006 on Figure 5? The DRE difference is even larger on that day, which should help identify differences in COT as the main cause.*
Agreed, the figure was changed.

---

## Referee Report (RR1)

**2nd Review of deGraaf, et al., 2019.**

This is my second review of this manuscript. I thank the authours for taking on-board many of the comments from my and the other review, and for improving the manuscript. It is easier to follow now, although there are still improvements that could be made to make it clearer (see the specific suggestions below, but there are likely more things that could be done). As well as these, a table showing the resolutions of the different instruments would be useful given the frequent re-gridding. Proof reading for grammar is also needed due to a few grammatical errors that hamper the reading somewhat.

**Specific suggestions**

p.2, L19 – "When these are accounted for, the remaining differences can be completely explained by the higher cloud optical thickness derived from POLDER compared to the other instruments. Additionally, a neglect of AOT at SWIR wavelengths in the method used for SCIAMACHY and OMI-MODIS accounts about a third of the difference between POLDER and OMI-MODIS DRE, which is mainly evident at high values of the aerosol DRE."

*- These two sentences don't seem to agree with each other. Perhaps the first should be modified to "the remaining differences can be almost completely explained", or similar.*

p.3, L56 – "The DRE from SCIAMACHY was compared to Hadley Centre Global Environmental Model version 2 (HadGEM2) climate model simulations, showing that even this GCM, which simulated a large warming over the south-east Atlantic, still fell short in simulating the UV-absorption by smoke (de Graaf et al., 2014)."

- *It would be good to quote the warming from the model, or the degree of underestimate.*

From the first review :- p.5, L2 – 'CER was derived from collocated MODIS measurements.' Would it not be better for POLDER to retrieve the CER? Is this retrieval not possible? Could MODIS CER be biased by the overlying aerosol, or by inhomogeneous clouds, etc.?

POLDER does not have measurement in the near infrared. MODIS CER is retrieved primarily from the 2.1_m channel over the ocean. It can potentially be biased by the presence of aerosols above clouds. However, in the region of interest, the aerosols typically observed above clouds (i.e. biomass burning aerosols) are characterised by a large Ångström exponent. Therefore, their contribution to the signal at 2.1_m is expected to be negligible. This is the same argument that is used for the (OMI-)MODIS retrievals, except at 1.2_m. At 2.1_m the effect will be much smaller. Regarding the 3D effect, several filters are used on the POLDER AAC products in order to reject inhomogeneous clouds (Waquet et al., 2013b, GRL).

*I was thinking of the direct retrieval of CER by POLDER using the separation of the peaks in the polarized scattering phase function (Breon, et al., 2005). This retrieval is considered to bypass some of the potential biases in visible+near-infrared based retrievals (e.g., MODIS, etc.). There are such POLDER CER datasets available, although they may be limited to regions with more homogeneous clouds. A comparisons in some regions should be possible. It may be beyond the scope of this study, but should be mentioned at least.*

p.5, L10 – "Such an estimate is often missing."

*Better as "Such estimates are rarely given in the literature.", in order to make it clearer that you are referring to previous studies.*

p.5, L11 - "Moreover, the correct characterization of the spectral properties of the overlying aerosols is circumvented by DAA."

*Not sure what you mean here. How is it circumvented? This doesn't seem to fit with the previous sentences.*

Section 3.1 – *You should quote the area average for OMI-MODIS when collocated to the POLDER grid (in the text and in Table 1) since this would give an idea of how much difference the restricted retrievals of POLDER makes and would give support to the statements in the text that this makes a big difference (rather than the larger individual values).*

Fig. 2 & Table 2– *In caption for (b) and in Table 2 (for second set of values) it would be better to say that OMI-MODIS and SCHIMACHY were re-gridded to a 6km grid and that averages are only calculated for the 6km gridpoints for which all 3 instruments produced a measurement.*

p.9, L24 – "Additionally, the sampling was checked by gridding the finer POLDER data to the coarser OMI grid"

*Please make it clear in the text how this was done – was an average over each OMI-MODIS grid-box taken, or was just one value sampled for each OMI-MODIS box? However, if the latter then this might make the statistics less robust. Doing an average of lots of POLDER values for each box would be better.*

p.9, L28 – "loose" -> "lose"

p.10, L38 – "SCIAMACHY DRE is very similar to OMI-MODIS DRE for all pixels sampled by these instruments (not shown)."

*This is a strange blanket statement that seems to suggest that SCIAMACY and OMI agree for all pixels, which is surely not the case. This needs to be more quantitative, or show some results (e.g, a scatter plot).*

Fig. 5a – *there are 4 lines for each group of COT values, but only two entries in the legend (reff=8 and 12).*

p.11 L21 – "Irrespective of AOT or COT, an error of 20% in COT can lead to an error in DRE of 50 Wm−2."

*This is written in a slightly odd way that makes it seem like the error is always 50 W/m2 no matter what the COT or AOT values are. In fact it looks like the error value ranges from about 30 to 80 W/m2. I suggest :-*

*"An error of 20% in COT leads to an error in DRE of between ~30 and 80 Wm−2 for the COT values tested (COT=8 and COT=12), irrespective of the AOT."*

p.11, L23 – "A note for DAA is in order here: because the simulated aerosol-free cloud spectrum is computed from the COT and CER retrieved from the measured aerosol-cloud spectrum, the spectra are the same in the SWIR, cancelling retrieval errors in the COT. A test with OMI-MODIS spectra and POLDER COT regridded to the OMI grid yielded erratic DRE, even though the POLDER COT retrieved in the visible is probably superior to the OMI-MODIS COT retrieved in the SWIR. However, in such a case a simulated cloud spectrum using POLDER COT is often different from the spectrum by MODIS in the SWIR, yielding aerosol effects even without aerosols. For the POLDER DRE calculation this effect is different, because the DRE is computed using the scene twice with the same retrieved COT."

*This is difficult to follow and understand – it could do with a rewrite and a more careful explanation of what you mean.*

p.11, L41 – "However, it is assumed to have negligible effect in the SWIR from about 1.2 µm, which may be an underestimation (in AOT)."

*This sentence doesn't really make sense. The first part of the sentence seems to say that AOT errors have a negligible effect, but does not say on what. The second part suggests there is an underestimation of AOT, but it is not clear.*

p.12, L26 – "Note that POLDER COT is retrieved at 0.87 µm, while COT from OMI-MODIS is retrieved at 1.2 µm, which effectively is the MODIS channel."

*MODIS has lots of channels, so I'm not sure I understand the last part of the sentence. Also, MODIS can retrieve COT using the 0.86um channel (as for POLDER), although combined with the 2.1um or 3.7um channel. However, most of the signal comes from the 0.86um channel. Or does the possibility of the above-cloud aerosol prevent the use of this?*

p.12, L38 – "Here, POLDER COT is regridded, while the MODIS radiances are averaged in the OMI footprint and one COT is retrieved for that OMI pixel."

*It's not clear at this point in the manuscript why this is being done. It is interesting to compare the effect of averaging the (1km resolution, or higher? Please state) MODIS radiances to the OMI resolution before doing the COT retrieval. But it needs some introduction at the start of the paragraph about the reasons for doing it. Is this is what is done to get the OMI-MODIS COT values used in the DRE retrieval? Or are the 1km MODIS COT values averaged over each OMI gridbox? It should be mentioned in Section 2.3 how this is done. If the MODIS COT values are averaged, you should also show a scatter plot of the POLDER vs OMI-MODIS COT values where the POLDER values are averaged to the OMI grid and the MODIS COT values are averaged. Also, please state what you do with the POLDER COT values. Do you average them over the OMI grid-boxes?*

Table 3 – *it would be good to add that the POLDER data is gridded to the coarser OMI-MODIS grid in the caption. And whether the POLDER data was averaged over the OMI boxes, or interpolated/sampled.*

p.13, L30 - "This approach removes issues related to selecting high positive DRE values by filtering on COT and CF, which introduce large differences in the average DRE. Even if the same filtering is used for the CF and COT for all instruments, different areas will be sampled, because the CF and COT retrieved by the different instruments may be different."

*My issue here from the 1st review was that it wasn't clear what filtering by CF and COT was done. Looking through the manuscript I see that this probably refers to the requirement of minimum CF and COT values for a DRE retrieval in Sections 2.1-2.3. However, since this information is in the methods section and not mentioned since, it wasn't clear here what was being referred to. Before talking about CF and COT filtering here it would be good to reiterate that different minimum CF and COT values were used for the different instruments.*

*Also, I think here you mean "This approach removes issues related to filtering based on COT and CF, which can select high positive DRE values and lead to large differences in the average DRE."*

p.13, L38 – "This difference was removed by gridding POLDER to the coarser OMI grid, improving the comparison between OMI-MODIS DRE and POLDER DRE."

*This wouldn't completely remove the effect because the effect of averaging the reflectances from the coarser resolution instrument (and then doing the retrieval) would still remain – this causes differences due to the non-linearity between the reflectances and retrieved product. I.e., it's not just a case of averaging the POLDER values to a coarser grid.*

p.13, L62 – "The underestimation of the AOT for high values can explain about a third of the difference in DRE between POLDER and OMI-MODIS om 12 August 2006, an overestimation of AOT by POLDER is difficult to establish."

*The last part of this sentence (after the comma) doesn't fit with the rest. Also "om 12 August".*

p.13, L69 – "Normally, MODIS COT retrievals at 0.8 and 2.1 μm  are close to POLDER COT for fully clouded scenes with liquid water clouds (Zeng et al., 2012) (not considering overlying smoke). However, to avoid biases from smoke absorption, the MODIS channels at 1.2 and 2.1 μm are used to derive COT and CER for OMI-MODIS DRE retrievals, which may further influence the results."

*- This information should also be in the methods section to explain why the usual MODIS channels are not used for COT. Also, the second "retrievals" word should be removed.*

p.13, L76 – "The difference between COT from OMI-MODIS and POLDER on 12 August 2006 can explain about 80% of the difference in DRE on that day."

*- Please state whether this is before or after the footprints have been collocated.*

p.13, L85 – "However, a test using this approach using DDA on OMI-MODIS spectra using POLDER COT yielded very erratic results." -> "However, a test of this approach using DDA on OMI-MODIS spectra but with POLDER COT yielded very erratic results." (too many instances of "using").

p. 13, L87 – "clouds spectrum" -> "cloud spectrum".

**References**

Bréon, F. M., & Doutriaux-Boucher, M. (2005). A comparison of cloud droplet radii measured from space. IEEE Transactions on Geoscience and Remote Sensing, 43(8), 1796–1805. https://doi.org/10.1109/TGRS.2005.852838

---

## Author Response (AR2)

*Reviewer #1*
**2nd Review of de Graaf, et al. 2019.**

This is my second review of this manuscript. I thank the authours for taking on-board many of the comments from my and the other review, and for improving the manuscript. It is easier to follow now, although there are still improvements that could be made to make it clearer (see the specific suggestions below, but there are likely more things that could be done). As well as these, a table showing the resolutions of the different instruments would be useful given the frequent re-gridding. Proof reading for grammar is also needed due to a few grammatical errors that hamper the reading somewhat.

The reviewer is thanked for the amount of time invested to improve the paper. All issues raised by the reviewer have been considered and addressed below. All changes to the manuscript are indicated for each answer. The desired table (see Table 1) was added to the new manuscript. The manuscript was carefully spell-checked and checked for grammar and errors by several co-authors.

Table 1: Spatial and temporal resolution of the different satellite instruments as used in this paper. Grid sizes of SCIAMACHY and OMI are those at nadir, grid sizes of POLDER and MODIS are fixed.

| Instrument | Platform | Local equator crossing time | Global coverage (days) | Pixel size (km × km) | Operation period |
|---|---|---|---|---|---|
| POLDER | PARASOL | 13:33 | 1 | $6 \times 6$ | 2004 – 2013 |
| SCIAMACHY | EnviSat | 10:00 | 6 | $60 \times 30$ | 2002 – 2012 |
| OMI | Aura | 13:38 | 1 | $13 \times 24$ | 2004 – present |
| MODIS | Aqua | 13:30 | 1 | $0.5 \times 0.5$ | 2002 – present |

**Specific suggestions**

p.2, L19 "When these are accounted for, the remaining differences can be completely explained by the higher cloud optical thickness derived from POLDER compared to the other instruments. Additionally, a neglect of AOT at SWIR wavelengths in the method used for SCIAMACHY and OMI- MODIS accounts about a third of the difference between POLDER and OMI-MODIS DRE, which is mainly evident at high values of the aerosol DRE."

– These two sentences don't seem to agree with each other. Perhaps the first should be modified to "the remaining differences can be almost completely explained", or similar.

The idea here is that the uncertainties in the cloud retrievals are already sufficient to explain the differences found between the DRE from the different instruments. Additionally, there are uncertainties in the AOT retrievals, which also contribute to the differences. Added together, the uncertainties are more than the differences found, so there are cancelling errors, or the error estimates are too large. The text was rephrased as follows:

When these are accounted for, the remaining differences can be explained by a higher cloud optical thickness (COT) derived from POLDER compared to the other instruments, and a neglect of AOT at SWIR wavelengths in the method used for SCIA-MACHY and OMI-MODIS. The higher COT from POLDER by itself can explain the difference found in DRE between POLDER and the other instruments. The AOT underestimation is mainly evident at high values of the aerosol DRE and accounts for about a third of the difference between POLDER and OMI-MODIS DRE.

p.3, L56 -" The DRE from SCIAMACHY was compared to Hadley Centre Global Environmental Model version 2 (HadGEM2) climate model simulations, showing that even this GCM, which simulated a large warming over the south-east Atlantic, still fell short in simulating the UV-absorption by smoke (de Graaf *et al.*, 2014)."

 - It would be good to quote the warming from the model, or the degree of underestimate.

Agreed. The following text was added, quoted from the paper: "Simulated monthly averaged aerosol DRE from HadGEM2 were a factor of 5 lower than SCIAMACHY observations.

From the first review: - p.5, L2 "CER was derived from collocated MODIS measurements."

Would it not be better for POLDER to retrieve the CER? Is this retrieval not possible? Could MODIS CER be biased by the overlying aerosol, or by inhomogeneous clouds, etc.?

*POLDER does not have measurement in the near infrared. MODIS CER is retrieved primarily from the 2.1µm channel over the ocean. It can potentially be biased by the presence of aerosols above clouds. However, in the region of interest, the aerosols typically observed above clouds (i.e. biomass burning aerosols) are characterised by a large Ångström exponent. Therefore, their contribution to the signal at 2.1µm is expected to be negligible. This is the same argument that is used for the (OMI-)MODIS retrievals, except at 1.2µm. At 2.1µm the effect will be much smaller. Regarding the 3D effect, several filters are used on the POLDER AAC products in order to reject inhomogeneous clouds (Waquet et al., 2013b, GRL).*

I was thinking of the direct retrieval of CER by POLDER using the separation of the peaks in the polarized scattering phase function (Breon, *et al.*, 2005). This retrieval is considered to bypass some of the potential biases in visible+near-infrared based retrievals (e.g., MODIS, etc.). There are such POLDER CER datasets available, although they may be limited to regions with more homogeneous clouds. A comparisons in some regions should be possible. It may be beyond the scope of this study, but should be mentioned at least.

In this paper the POLDER DRE data that are described in Peers *et al.* (2015) are compared with the other datasets. In this version of the data the cloud droplet sizes come from MODIS measurements as mentioned above. An application of CER by POLDER as the reviewer suggests is certainly interesting, and could be considered for a new version of the DRE retrieval.

A note was added to the text:
"CER may be retrieved directly by POLDER for specific cases using the separation of the peaks in the polarized scattering phase function (Bréon *et al.*, 2015) , but here we use the data as described in Peers *et al.* (2015)."
And the reference was added.

p.5, L10 "Such an estimate is often missing." Better as "Such estimates are rarely given in the literature.", in order to make it clearer that you are referring to previous studies.
Agreed, the text was changed accordingly.

p.5, L11 - Moreover, the correct characterization of the spectral properties of the overlying aerosols is circumvented by DAA." Not sure what you mean here. How is it circumvented? This doesn't seem to fit with the previous sentences.
This is a correct observation. It doesn't fit the previous statement, and should be a separate statement. DAA circumvents any aerosol property assumptions by using the spectral measurements instead of running a model twice (once with and once without (modelled) aerosols). A new paragraph was started and the text was changed as follows:
"The dependence on uncertanties in the spectral properties of the overlying aerosols is small by DAA, because in this method the spectral measurements are used, not a model."

Section 3.1 -You should quote the area average for OMI-MODIS when collocated to the POLDER grid (in the text and in Table 1) since this would give an idea of how much difference the restricted retrievals of POLDER makes and would give support to the statements in the text that this makes a big difference (rather than the larger individual values).
The desired information is available from Table 3 and Figure 2, which can be directly compared to Table 1. The introductory section 3.1 discusses the two selected cases before any analysis, using the data as they are, showing a large difference on one day and more moderate on the other. To quote here values from the regridded cases would confuse readers, since the regridding is not mentioned up to this point. Therefore, these values are discussed at the end of section 3.2.2.

Fig. 2 & Table 2 -In caption for (b) and in Table 2 (for second set of values) it would be better to say that OMI-MODIS and SCHIMACHY were re-gridded to a 6km grid and that averages are only calculated for the 6km grid points for which all 3 instruments produced a measurement.
Agreed. The caption was changed as follows:
"b) Same as a), but for OMI-MODIS and SCIAMACHY pixels that were regridded to the $6 \times 6$ km$^2$ POLDER grid. Averaged values were only calculated from grid points that were covered by all three instruments. The number of collocated pixels that are covered by all three instruments is given in the lower panel in b)."

p.9, L24 -"Additionally, the sampling was checked by gridding the finer POLDER data to the coarser OMI grid" Please make it clear in the text how this was done. Was an

average over each OMI-MODIS grid-box taken, or was just one value sampled for each OMI-MODIS box? However, if the latter then this might make the statistics less robust. Doing an average of lots of POLDER values for each box would be better.

The smaller POLDER pixels were sampled over the OMI footprint using an advanced 2D Gaussian weighting function, in the same way as MODIS pixels are weighted in the OMI footprint for the OMI-MODIS DRE computation. This is indeed much more accurate than a simple nearest neighbour sampling. The following was added to the text:

"The smaller POLDER pixels were averaged over the OMI footprint using a 2D Gaussian weighting function. This procedure is exactly the same for the averaging of MODIS pixels in an OMI footprint in the OMI-MODIS DRE computation, and described in detail in De Graaf *et al.* (2016)."

p.9, L28 " "loose" − > "lose"
Indeed. Thank you.

p.10, L38 " "SCIAMACHY DRE is very similar to OMI-MODIS DRE for all pixels sampled by these instruments (not shown)." This is a strange blanket statement that seems to suggest that SCIAMACY and OMI agree for all pixels, which is surely not the case. This needs to be more quantitative, or show some results (e.g, a scatter plot).
Agreed, the sentence on SCIAMACHY data doesn't really add anything to this paragraph and was removed.

Fig. 5a " there are 4 lines for each group of COT values, but only two entries in the legend (reff=8 and 12).
The legend shows that each group of two values refer to either 10° or 60° viewing zenith angle. The legend in the figure was changed to show also this color.

p.11 L21 -"Irrespective of AOT or COT, an error of 20% in COT can lead to an error in DRE of 50 Wm"-2." This is written in a slightly odd way that makes it seem like the error is always 50 W/m2 no matter what the COT or AOT values are. In fact it looks like the error value ranges from about 30 to 80 W/m2. I suggest :- "An error of 20% in COT leads to an error in DRE of between 30 and 80 Wm-2 for the COT values tested (COT=8 and COT=12), irrespective of the AOT."
The reviewer is correct that this is not for any COT and AOT value, but only those that are shown. However, the error range that the reviewer suggest is far too large. Therefore, the text was changed as follows:

An error of 20% in COT can lead to an error in DRE of *about* 50 Wm$^{-2}$ , for COT in the range of 8–16, irrespective of the AOT.

p.11, L23 " "A note for DAA is in order here: because the simulated aerosol-free cloud spectrum is computed from the COT and CER retrieved from the measured aerosol-cloud spectrum, the spectra are the same in the SWIR, cancelling retrieval errors in the COT. A test with OMI-MODIS spectra and POLDER COT regridded to the OMI grid

yielded erratic DRE, even though the POLDER COT retrieved in the visible is probably superior to the OMI-MODISCOT retrieved in the SWIR. However, in such a case a simulated cloud spectrum using POLDER COT is often different from the spectrum by MODIS in the SWIR, yielding aerosol effects even without aerosols. For the POLDER DRE calculation this effect is different, because the DRE is computed using the scene twice with the same retrieved COT."

This is difficult to follow and understand " it could do with a rewrite and a more careful explanation of what you mean.
The text was rewritten as follows:
The DRE is computed from the difference between a measured and simulated spectrum, which both have exactly the same COT and CER (since the simulation is done with the COT and CER retrieved from the measured spectrum). Therefore, any errors in the COT and CER retrieval have no influence on the difference between the two spectra and do not show in the DRE. However, if the COT or CER for the simulation were taken from a different measurement, however accurate, the simulated and measured spectra may be very different, giving rise to large DRE values, even without overlying aerosols. This was observed in a test where POLDER COT, regridded to the OMI grid, was used in the DRE computation, instead of the COT from the OMI-MODIS spectra. Even though the POLDER COT was probably more accurate than the OMI-MODIS COT, the derived DRE was very erratic.

p.11, L41 " "However, it is assumed to have negligible effect in the SWIR from about 1.2 $\mu$m, which may be an underestimation (in AOT)."

This sentence doesn't really make sense. The first part of the sentence seems to say that AOT errors have a negligible effect, but does not say on what. The second part suggests there is an underestimation of AOT, but it is not clear.
Agreed. The text was changed as follows:
"However, the AOT of small particles like smoke is assumed to be negligible in the SWIR from about 1.2 $\mu$m, which may be an underestimation.

p.12, L26 -"Note that POLDER COT is retrieved at 0.87 $\mu$m, while COT from OMI-MODIS is retrieved at 1.2 $\mu$m, which effectively is the MODIS channel."

MODIS has lots of channels, so I'm not sure I understand the last part of the sentence. Also, MODIS can retrieve COT using the 0.86um channel (as for POLDER), although combined with the 2.1 um or 3.7 um channel. However, most of the signal comes from the 0.86um channel. Or does the possibility of the above-cloud aerosol prevent the use of this?
"MODIS channel" was changed to "MODIS measurement".
Indeed, the whole idea of retrieving COT in the SWIR is to avoid biases due to aerosol absorption in the UV-VIS channels.

p.12, L38 " "Here, POLDER COT is regridded, while the MODIS radiances are averaged in the OMI footprint and one COT is retrieved for that OMI pixel."

It's not clear at this point in the manuscript why this is being done.

It is strange to read this remark, when the reason for doing this was that the reviewer suggested this in the first review. In the first draft the OMI-MODIS values were re-sampled to the POLDER grid, in the current manuscript the gridding is to the coarser OMI grid, which is more accurate, but does not yield fundamentally different results. However, "Here, POLDER COT is regridded," was changed to "Here, POLDER COT values were averaged over the OMI footprint,"

It is interesting to compare the effect of averaging the (1km resolution, or higher? Please state) MODIS radiances to the OMI resolution before doing the COT retrieval. But it needs some introduction at the start of the paragraph about the reasons for doing it. Is this is what is done to get the OMI-MODIS COT values used in the DRE retrieval?

MODIS L1B reflectances have a spatial resolution of 250 or 500 m, depending on the channel. They have to be first co-added to 500 m and then gridded. The gridding over the OMI footprint is done using a 2D weighting function over the OMI footprint, de-scribed in De Graaf *et al.* (2016) and De Graaf *et al.* (2019).

Or are the 1km MODIS COT values averaged over each OMI gridbox? It should be mentioned in Section 2.3 how this is done.

No, this is not done. In section 3.2.2 it states: "for MODIS the radiances were averaged and one COT is retrieved for that OMI pixel."

If the MODIS COT values are averaged, you should also show a scatter plot of the POLDER vs OMI -MODIS COT values where the POLDER values are averaged to the OMI grid and the MODIS COT values are averaged.

But this is not done.

Also, please state what you do with the POLDER COT values. Do you average them over the OMI grid -boxes?

Yes, as mentioned previously, the POLDER values are not taken in a nearest neighbour fashion, but properly averaged using a weighting function over the OMI footprint, and described in De Graaf *et al.* (2016) and De Graaf *et al.* (2019). This was added to the manuscript.

Table 3 " it would be good to add that the POLDER data is gridded to the coarser OMI-MODIS grid in the caption. And whether the POLDER data was averaged over the OMI boxes, or interpolated/sampled.

This is already explained in the text. In the table are three headers: native grid, col-located POLDER grid, collocated OMI grid, which should make it clear which values belong to which analysis.

p.13, L30 -"This approach removes issues related to selecting high positive DRE val-ues by filtering on COT and CF, which introduce large differences in the average DRE. Even if the same filtering is used for the CF and COT for all instruments, different areas will be sampled, because the CF and COT retrieved by the different instruments may

be different."

My issue here from the 1st review was that it wasn't clear what filtering by CF and COT was done. Looking through the manuscript I see that this probably refers to the requirement of minimum CF and COT values for a DRE retrieval in Sections 2.1-2.3. However, since this information is in the methods section and not mentioned since, it wasn't clear here what was being referred to. Before talking about CF and COT filtering here it would be good to reiterate that different minimum CF and COT values were used for the different instruments. Also, I think here you mean "This approach removes issues related to filtering based on COT and CF, which can select high positive DRE values and lead to large differences in the average DRE."

The suggestion of the reviewer was adopted and "This approach removes issues related to selecting high positive DRE values by filtering on COT and CF, which introduce large differences in the average DRE." was changed to "This approach removes issues related to filtering based on COT and CF, which can select high positive DRE values and lead to large differences in the average DRE." The filter settings are mentioned in the methods section under the different instrument descriptions, which seems the correct place for this information and thus not changed.

p.13, L38 " "This difference was removed by gridding POLDER to the coarser OMI grid, improving the comparison between OMI-MODIS DRE and POLDER DRE."
This wouldn't completely remove the effect because the effect of averaging the reflectances from the coarser resolution instrument (and then doing the retrieval) would still remain " this causes differences due to the non-linearity between the reflectances and retrieved product. I.e., it's not just a case of averaging the POLDER values to a coarser grid.

Agreed; "removed" was changed to "reduced".

p.13, L62 " "The underestimation of the AOT for high values can explain about a third of the difference in DRE between POLDER and OMI-MODIS om 12 August 2006, an overestimation of AOT by POLDER is difficult to establish."

The last part of this sentence (after the 1comma) doesn 't fit with the rest. Also "om 12 August".

Agreed. Changed to:

"The underestimation of the AOT for high values can explain about a third of the difference in DRE between POLDER and OMI-MODIS on 12 August 2006. POLDER AOT may be overestimated, but this is difficult to quantify."

p.13, L69 " "Normally, MODIS COT retrievals at 0.8 and 2.1 $\mu$m retrievals are close to POLDER COT for fully clouded scenes with liquid water clouds (Zeng *et al.*, 2012) (not considering overlying smoke). However, to avoid biases from smoke absorption, the MODIS channels at 1.2 and 2.1 $\mu$m are used to derive COT and CER for OMI-MODIS DRE retrievals, which may further influence the results."

- This information should also be in the methods section to explain why the usual

MODIS channels are not used for COT. Also, the second "retrievals" word should be removed.

"retrievals" was removed. The method mentions: "Cloud properties were determined at 1.2 and 1.6 $\mu$m, where absorption by smoke is assumed to be negligible. "

p.13, L76 " "The difference between COT from OMI-MODIS and POLDER on 12 August 2006 can explain about 80% of the difference in DRE on that day."

- Please state whether this is before or after the footprints have been collocated.

This is after sampling issues have been removed. This is described in the text and the conclusion section, because that states that sampling issues are the largest contribution to the differences and should be removed first.

[revised manuscript text omitted]